# Surface chemical heterogeneous distribution in over-lithiated Li$_{1+x}$CoO$_2$ electrodes

Gang Sun[1,2,9], Fu-Da Yu[3,9], Mi Lu[4], Qingjun Zhu[1], Yunshan Jiang[5], Yongzhi Mao[5], John A. McLeod[6], Jason Maley[7], Jian Wang ●[8,10] ✉, Jigang Zhou ●[8,10] ✉ & Zhenbo Wang ●[1,2,5,10] ✉

In commercial Li-ion batteries, the internal short circuits or over-lithiation often cause structural transformation in electrodes and may lead to safety risks. Herein, we investigate the over-discharged mechanism of LiCoO$_2$/graphite pouch cells, especially spatially resolving the morphological, surface phase, and local electronic structure of LiCoO$_2$ electrode. With synchrotron-based X-ray techniques and Raman mapping, together with spectroscopy simulations, we demonstrate that over-lithiation reaction is a surface effect, accompanied by Co reduction and surface structure transformation to Li$_2$CoO$_2$/Co$_3$O$_4$/CoO/Li$_2$O-like phases. This surface chemical distribution variation is relevant to the depth and exposed crystalline planes of LiCoO$_2$ particles, and the distribution of binder/conductive additives. Theoretical calculations confirm that Li$_2$CoO$_2$-phase has lower electronic/ionic conductivity than LiCoO$_2$-phase, further revealing the critical effect of distribution of conductive additives on the surface chemical heterogeneity evolution. Our findings on such surface phenomena are non-trivial and highlight the capability of synchrotron-based X-ray techniques for studying the spatial chemical phase heterogeneity.

As the vigorous development of "3C products" (Computer, Communication, and Consumer Electronics) and the rapid expansion of the energy storage and electric vehicle markets, the application of lithium-ion batteries (LIBs) has experienced explosive growth[1,2]. As a dominant cathode material, LiCoO$_2$ is widely used in commercial Li-ion batteries due to its superior volumetric energy density and stability[3,4]. Nowadays, research on LIBs are mainly focusing on improving their energy density or exploring their capacity fade mechanisms. Many efforts have been devoted to exploring the structural transitions and capacity fading mechanisms of LiCoO$_2$-based batteries at high voltages to

improve the energy density of LIBs[5–8]. However, the higher the energy density the LIB possesses, the higher the corresponding safety risks.

Due to the inconsistency among the battery cells, over-lithiation can easily occur in LIB packs. For an individual cell, overuse and the differences in internal local structures or contacts can also lead to over-discharge. In the cases of internal short cricuits, over-discharge, or other abuse conditions of LIBs, the cathode/anode/separators/current collector of the full battery system may undergo significant degradation, thus giving rise to severe safety issues[9]. Therefore, it is necessary to understand the working mechanisms of the electrode

[1]College of Materials Science and Engineering, Shenzhen University, 518071 Shenzhen, China. [2]College of Physics and Optoelectronic Engineering, Shenzhen University, 518060 Shenzhen, China. [3]College of Material Science and Engineering, Huaqiao University, 361021 Xiamen, China. [4]Key Laboratory of Functional Materials and Applications of Fujian Province, School of Materials Science and Engineering, Xiamen University of Technology, 361024 Xiamen, China. [5]School of Chemistry and Chemical Engineering, Harbin Institute of Technology, 150001 Harbin, China. [6]Department of Electrical & Computer Engineering, Western University, London, ON N6A 5B9, Canada. [7]Department of Chemistry and Saskatchewan Structural Sciences Centre, University of Saskatchewan, Saskatoon, SK S7N 5C9, Canada. [8]Canadian Light Source Inc., University of Saskatchewan, Saskatoon, SK S7N 2V3, Canada. [9]These authors contributed equally: Gang Sun, Fu-Da Yu. [10]These authors jointly supervised this work: Jian Wang, Jigang Zhou, Zhenbo Wang. ✉e-mail: jian.wang@lightsource.ca; jigang.zhou@lightsource.ca; wangzhb@hit.edu.cn

material in the over-discharge state for broadening the battery operation tolerance, especially for safety. Previous reports have been demonstrated that over-discharge can significantly influence the anode[10–12]. Over-discharge will cause a voltage increase at the anode and serious dissolution of the copper current collector, degrading the electrochemical performance of LIBs[13–15]. Over-stoichiometric $Li^+$ will also be intercalated into the cathode during the over-discharge process, and make irreversible changes to the local structure, accelerating the battery degradation[16]. Crompton et al.[9,17] declared that $Li^+$ over-insertion degrades the cathode material, which could be the primary degradation mechanism of batteries, and modifying the cathode to stabilize it against the near-zero volt storage conditions may further improve near-zero voltage storage tolerance of LIBs. Shu et al.[18] carried out a comparative study to reveal the effect of over-discharged cathode materials ($LiFePO_4$, $LiNiO_2$, and $LiMn_2O_4$) and discovered that deep over-discharge has severely harmful impacts on most cathode materials.

However, recent results reveal that the impact of over-discharge is highly dependent on the local structures of the cathode: if the cathode material can safely and reversibly accommodate excess lithium, over-discharge can even be beneficial for battery performance. Over-lithiating the cathode before the first cycle offers the promise of retaining full battery capacity if the cathode material was specifically prepared to safely accommodate the excess lithium[19]. Improved performance by over-discharging was also demonstrated in $Li_3V_2O_5$ and $Li_3Nb_2O_5$ cathodes, which adopt a crystal structure that has plenty of vacancies to accommodate excess lithium[20,21]. On the other hand, over-discharging $LiMn_2O_4$ was shown to significantly degrade performance, as the $M_3O_4$ spinel structure does not easily accommodate excess metal ions[22]. In particular, the battery performance of Li- and Mg-doped $LiCoO_2$ was shown to be resilient to over-discharge compared to native $LiCoO_2$[23], however the location of the excess Li in $LiCoO_2$ was not determined. These findings make it evident that detailed probes of electrode local structures are needed to elucidate the process of over-discharge.

Additionally, the over-lithiation behavior of $LiCoO_2$ has also been revealed by the traditional characterization methods (such as charge-discharge cycling, XRD, XPS, Raman, and HRTEM)[24], and found that reduction mechanisms of $LiCoO_2$ with Li are associated with the irreversible formation of metastable phase $Li_{1+x}CoO_{2-y}$, and then the final products of $Li_2O$ and Co metal. Robert et al.[10] analyzed the local structure evolution of $LiNi_{0.80}Co_{0.15}Al_{0.05}O_2$ with further lithiation by ex-XRD. The results indicated that the further lithiation allows the accommodation of an additional $Li^+$ into the host lattice and promotes the $R\bar{3}m$ $LiMO_2$ to $P\bar{3}m1$ $Li_2MO_2$ phase transformation. However, despite tremendous efforts and significant achievements, the degradation mechanisms in terms of defect chemistry and defect electronic structure of over-discharged cathode active materials (no matter in bulk or surface regions), especially in real batteries at high spatial resolution, are still not well established. The absence of local chemical spatial information in electrode characterization using XPS[16] and XANES[25] might be insufficient to provide a full understanding of the role of surface coupling effects in metal and oxygen sites (structure and redox) in affecting the cathode's stability or reversibility.

The study of the surface phenomena is non-trivial and limited by the scarcity of the suitable characterization tools, for example, transmission electron microscope (TEM) suffers from radiation damage and a limited statistics capability. Powerful synchrotron-based scanning transmission X-ray microscopy (STXM) and X-ray photoemission electron microscopy (X-PEEM) have been proved useful for characterizing bulk and surface chemical composition, electronic structure and conductivity variations on different crystalline facets, as well as revealing the primary causes of different reactions and stabilities of the crystalline facets[26–32]. X-PEEM, with the ultra-high vacuum compatibility and full-field imaging capabilities, can overcome the limitations

of conventional STXM in the soft X-ray energy range; as the latter can only image thin sections or fine particles with the thickness/size of tens to hundreds of nanometers to obtain sample bulk information[33,34].

In this work, X-PEEM, XANES, and Raman imaging analyses of the interface in the discharged/over-discharged $LiCoO_2$ composite electrodes in a commercial $LiCoO_2$/graphite pouch cell have been performed to gain an understanding of the over-discharge mechanisms, especially on the surface heterogeneity in terms of the morphological, surface phase structure, local electronic environment, element valence state, their interplay and the additive effects. Combining with theoretical calculations, we demonstrate that $Li_2CoO_2$/$Li_2O$/$Co_3O_4$/$CoO$-like phases (exclusion of $Li_{1+x}CoO_{2-y}$) are non-uniformly distributed in the surface of over-discharged $LiCoO_2$ particles, and their distribution variation is relevant to the size, depth and exposed crystalline planes of $LiCoO_2$, and the distribution of binder/conductive agents. The spatial distribution of chemical heterogeneity, morphological degradation, and an unanticipated Co-containing compound phase in the surface of over-discharged $LiCoO_2$ electrodes are further explored. In addition, it is expected that systematic and thorough studies of local chemical spatial information under abuse operations will provide other insights to develop advanced and safe electrode materials for LIBs, and guide the development of batteries with high tolerance.

## Results

### Chemical and structural evolution of discharged $LiCoO_2$

Here, the commercial $LiCoO_2$ composite electrode in a $LiCoO_2$/graphite pouch cell was used to study the over-discharge degradation mechanisms under different cut-off voltages. Detailed descriptions of the electrochemical performance of the $LiCoO_2$/graphite pouch cell can be found in the Methods section[33,35]. The charging-discharging plots are displayed in Fig. 1a. The pouch cell was charged/discharged at a voltage range of 3.0–4.35 V, which results in the standard charging product ($Li_{0.5}CoO_2$) and discharging product ($LiCoO_2$). When the cell was gradually deep-discharged to 0 V, the discharge curve is clearly divided into two stages, as shown in the enlarged view of Fig. 1a. The first stage shows a sharp drop in the voltage plot with a slow capacity increase, which is attributed to over-stoichiometric $Li^+$ intercalated into the lattice of $LiCO_2$ and transformed into $Li_{1+x}CoO_2$ (marked with a red arrow). Then, there follows a slow voltage drop accompanied by a larger capacity release, which is caused by the dissolution of the copper current collector triggered by an increase in the anode potential[9]. The Scanning Electron Microscope/Energy Dispersive X-ray (SEM/EDS) mapping in Fig. S1 illustrates the uniform distribution of Cu on the over-discharged $LiCoO_2$ electrode (Named D-0.0 V in Fig. 1a), which proves the above-mentioned dissolution phenomenon of the copper current collector. The local electronic structure and chemical phase components in the discharged $LiCoO_2$ electrode (D-3.0 V, as marked in Fig. 1a) and its evolution upon over-discharge (D-0.0 V) have been studied by X-ray absorption near edge structure (XANES) of the Co L-edge in Fig. 1b and O K-edge in Fig. 1c. The standard Co L-edge spectra of CoO and $LiCoO_2$ are also displayed in Fig. 1b as reference. An obvious phase change can be observed upon over-discharging based on the comparison of the high-quality Co $L_3$-edge spectrum of the deep-discharged D-0.0 V electrode to those of the D-3.0 V electrode and $LiCoO_2$ in Fig. 1b. The Co $L_3$-edge spectrum of the D-3.0 V electrode is very similar to that of $LiCoO_2$, indicating the same chemical phase remaining in the structure, which can also be proven by the O K-edge spectra of the D-3.0 V electrode and $LiCoO_2$ in Figs. 1c and S2f. The main peak (located at 781 eV) of the D-0.0 V electrode with a relatively higher intensity shoulder (located at 779 eV) is broadened compared with that of other samples, indicating the co-existence of $Co^{2+}$ and $Co^{3+}$ in the structure. The Co 2p X-ray photoelectron spectroscopy (XPS) spectra of D-3.0 V and D-0.0 V electrodes were used to detect the changes in chemical compositions in Figure S2. Compared with that of the D-3.0 V electrode (29.2% for $Co^{2+}$), a large amount of

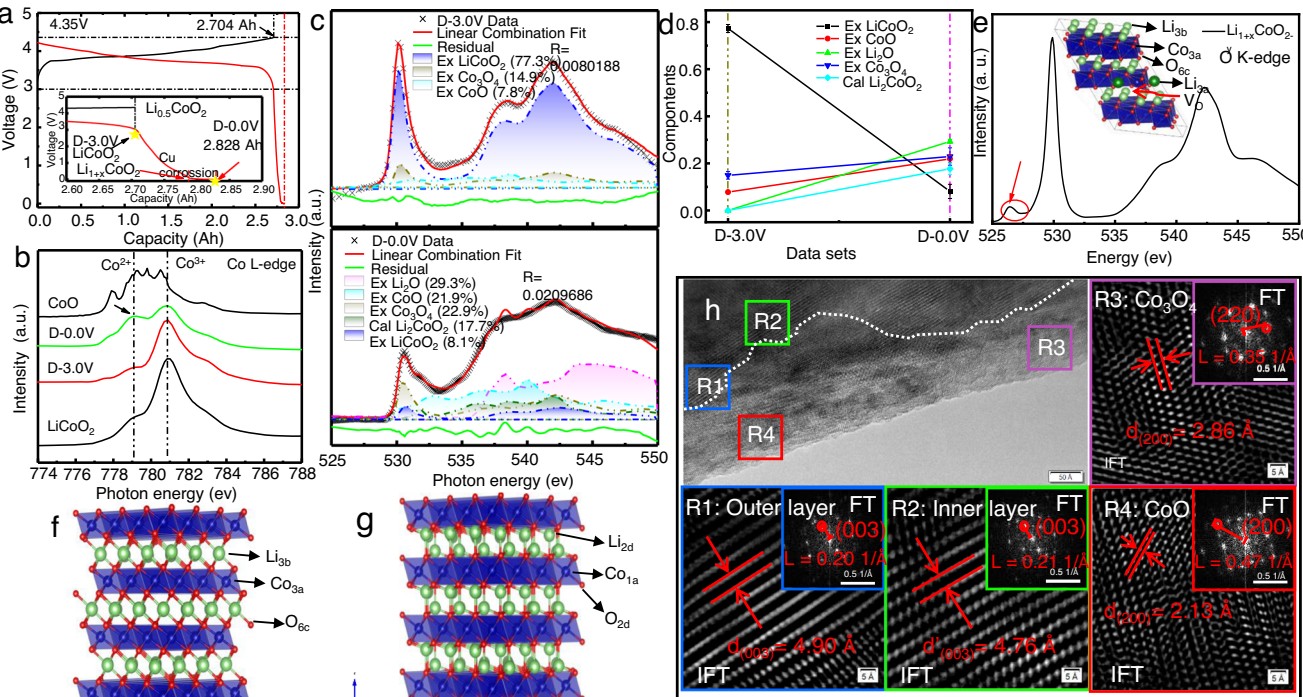

**Fig. 1 | XANES and HRTEM characterization of LiCoO₂ electrodes. a** The charge-discharge plot of LiCoO₂, with the inset showing the expanded capacity range from 2.60 to 2.90 Ah; **b** Co L₃-edge XANES spectra of CoO, D-0.0 V, D-3.0 V, and LiCoO₂ samples. **c** Linear combination fit of O K-edge XANES of D-3.0 V and D-0.0 V samples using experimental CoO, Li₂O, Co₃O₄, and LiCoO₂ spectra, and the calculated Li₂CoO₂ spectrum. **d** Comparison of the fitting components between D-3.0 V and D-0.0 V samples. **e** The calculated O K-edge XANES and the geometrical configuration (inset) of Li₁₊ₓCoO₂₋ᵧ. **f, g** The geometrical configurations of LiCoO₂ and Li₂CoO₂. **h** HRTEM, corresponding Fourier Transform (FT)/Inverse Fourier Transform (IFT) images (R1, R2, R3 and R4 region) of D-0.0 V electrode.

Co²⁺ (62.8%) was found on the surface of the D-0.0 V electrode, confirming the surface Co reduction during overdischarge, which is highly consistent with the conclusions of XANES spectroscopy. This is due to the over-stoichiometric Li⁺ being intercalated into the lattice structure of LiCoO₂ in the surface or near surface during the deep discharge, which induces Co reduction and the structural transformation to Li₁₊ₓCoO₂/Co₃O₄/CoO/Li₂O (The proof is as follows).

The O K-edge XANES spectrum of the D-0.0 V electrode shows some difference compared to that of the D-3.0 V electrode and the LiCoO₂ reference, which displays the reduced π* (~530 eV) intensity relative to that of the σ* (536-542 eV), as seen in Figs. 1c and S2f. The O pre-edge features between 529 and 532 eV reflect the hybridization of O 2p and Co 3d orbitals, and the main peaks in the range of 534-552 eV are due to transitions into O 2p hybridized with Co 4sp states. Therefore, the valence states of Co play a key role in the local electronic structure of O. To better understand the structure of LiCoO₂ electrodes under different discharge states, the experimental and calculated O K-edge XANES of Li₂CO₃, CoO, Li₂O, Co₃O₄, and LiCoO₂ are provided for comparison in Fig. S2. In addition, since Li₂CoO₂ and Li₁₊ₓCoO₂₋ᵧ have no real materials as a reference, only calculated O K-edge XANES can be provided (Figs. S2 and S3). Theoretical calculations were performed for the O K-edge using WIEN2k[36], a full-potential, all-electron density functional theory (DFT) code, and details are described in the supporting information. The geometrical configurations of LiCoO₂ and Li₂CoO₂ are displayed in Fig. 1f, g, respectively. The calculated and experimental data O K-edge XANES spectra are shown in Figs. 1c and S2 and S3. The linear combination fits of the O K-edge XANES spectra of D-3.0 V and D-0.0 V samples were performed using the experimental CoO, Li₂O, Co₃O₄, LiCoO₂ spectra, and the calculated Li₂CoO₂ spectrum, with only energy shifting and intensity scaling of each component while keeping the overall spectra shape to yield the best fitting correspondence. The fit yielded LiCoO₂, Co₃O₄, and CoO contributions of 77.3, 14.9, and 7.8% to the O K-edge XANES of D-3.0 V,

respectively. The fit yielded LiCoO₂, Co₃O₄, CoO, Li₂O, and Li₂CoO₂ contributions of 8.1, 22.9, 21.9, 29.3, and 17.7% to the O K-edge XANES of D-0.0 V, respectively. The comparison of the fitting components is displayed in Fig. 1d. The decrease in the contribution of LiCoO₂ indicates the structure evolution of the electrode during overdischarge (structure transition from LiCoO₂ + Li⁺ to Co₃O₄, CoO, Li₂O, and Li₂CoO₂). Previous reports provided a chemical formula close to [Li]ᵢₙₜₑᵣₛₗₐᵦ[Co₁₋ₓLiₓ]ₛₗₐᵦ[O₂₋ᵧ] to explain the phenomenon of Li-over-stoichiometric[37,38]. Herein, the Li₁₃Co₁₁O₂₃ model (inset of Fig. 1e) with substituted Li at the Co site and removed adjacent O was created to simulate the structure of Li₁₊ₓCoO₂₋ᵧ to calculate the O K-edge XANES spectra, and more details are described in the supporting document for Fig. S4. Unfortunately, the calculated Li₁₊ₓCoO₂₋ᵧ spectra do not agree well with the measured over-discharge D-0.0 V electrode data, as the measurements show a reduction in the pre-edge peak, not an enhancement in the features at even lower energies (red arrow in Fig. 1e). To sum up, the over-discharged LiCoO₂ electrode has Co reduction, and XANES along with XANES simulations indicates that CoO, Li₂O, Li₂CoO₂, and Co₃O₄-like phases (exclusion of LiₓCoO₂₋ᵧ) exist in the over-discharged LiCoO₂ electrode.

In addition, HRTEM and in-situ synchrotron XRD (sXRD) analyses were performed to investigate the surface chemical/structural evolution of LiCoO₂ electrodes during overdischarge, and the results are shown in Figs 1h and S5 and S6. The HRTEM and corresponding Fourier Transform (FT)/Inverse Fourier Transform (IFT) images clearly indicate that the existence of the cubic CoO/Co₃O₄/Li₂O phase (Figs. 1h and S5). Further analysis is performed on the 4 regions in Fig. 1h: R1 represents the outer layered structure; R2 represents the inner layered structure; R3 represents the Co₃O₄ region; R4 represents the CoO region. The lattice fringes of a representative layer structure with a d-spacing of 4.90 Å in R1 and 4.76 Å in R2 could be assigned to the (003) plane of layered LiCoO₂. The d₍₀₀₃₎-spacing in outer layered structure R1 region is greater than that in the R2 region, which may be due to the expansion

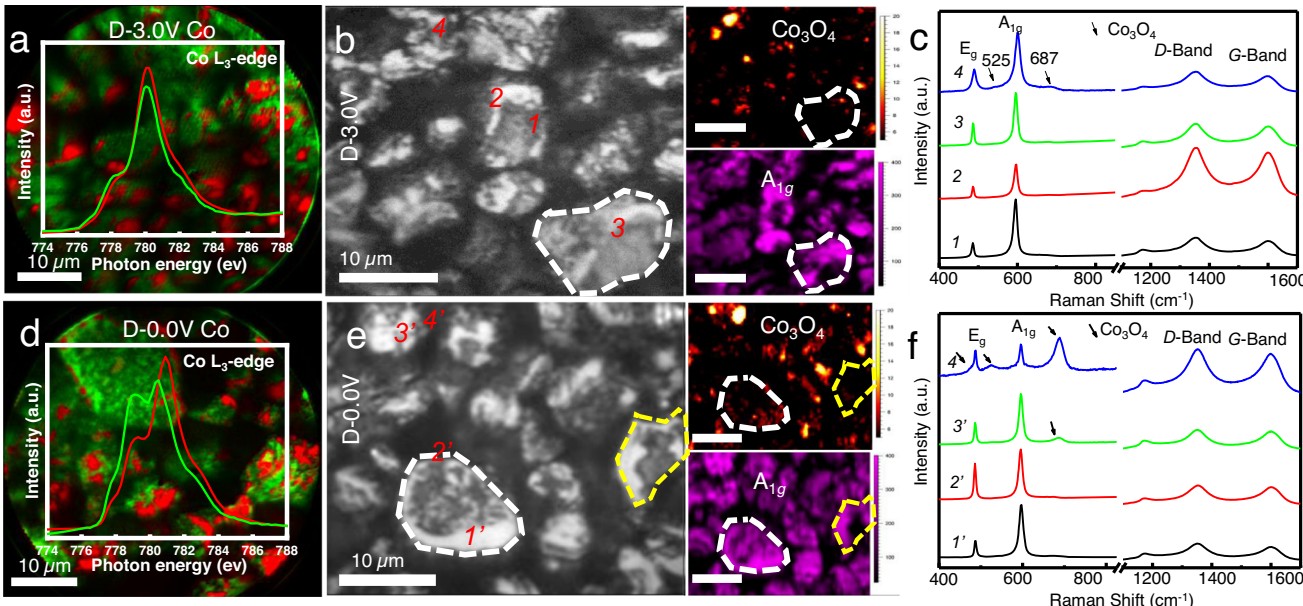

**Fig. 2 | X-PEEM and Raman chemical imaging of discharged LiCoO₂ electrodes. a, d** X-PEEM Co chemical mapping based on fitting Co L₃-edge image stacks (illustrations are Co L₃-edge XANES of red and green regions) of the electrode at two SOC states. **b, e** Bright-field reflection image (right) and Raman mapping of Co₃O₄ (687 cm⁻¹ peak intensity) and LiCoO₂ (A₁g peak intensity) (left) of the electrode at two SOC states. **c, f** Raman spectra extracted at selected spots on (**b, d**). **a–c** are for the D-3.0 V electrode; **d–f** are for the D-0.0 V electrode.

of the interlayer spacing caused by the intercalation of over-stoichiometric Li⁺ ions into the Li layer during the overlithiation process (the layered LiCoO₂ transformed to the layered Li₂CoO₂). The lattice fringe with a $d$-spacing of 2.86 Å in R3 region could be assigned to the (220) plane of spinel Co₃O₄ (JCPDS file nos. 43–1003, $a$ = 8.084 Å). It is worth noting that the R4 region exhibits a clear lattice fringe with a $d$-spacing of 2.13 Å, corresponding to the (200) plane of CoO (JCPDS file nos. 78–0431, $a$ = 4.2667 Å)[39]. In addition, for the surface of the D-0.0 V electrode a thin coating with thickness 2–4 nm was bound to the surface of the LiCoO₂ structure and distributed in regions in Fig. S5. It was confirmed from the enlarged area and the FT images that the outer layer in Fig. S5 is Li₂O phase, in which a $d$-spacing of 0.263 nm corresponds to the (111) plane of Li₂O[40]. Furthermore, in-situ synchrotron XRD of a LiCoO₂/graphite pouch cell was also used to investigate the structural evolution of the LiCoO₂ electrode during overdischarge from 3.0 to 0.0 V at 20 mA g⁻¹, and the result is shown in Fig. S6. All the diffraction peaks have no significant change (the position of the main peak) and no impurity peaks appear (such as CoO/Co₃O₄/Li₂O phase), only a slight difference in intensity, which indicates that overlithiation is only a surface phenomenon and has little effect on the bulk structure. However, a small enhancement of (006) peak intensity of the LiCoO₂ phase can be observed during the overdischarge process, while the (012) peak intensity has no intensity change (insets of Fig. S6a). All synchrotron XRD data were collected under the same conditions, so the intensity change in the diffraction peak could only be due to introduction of atom(s) in the unit cell, which resulted in a change in the structure factor. So, the (006) peak increase may indicate excessive intercalation of Li⁺ in the Li layer, which in turn produces the Li₂CoO₂ structure, while the influence on other directions is small. It can be concluded that the formation of Li₂CoO₂/Li₂O/Co₃O₄/CoO-like phases upon overlithiation of LiCoO₂ (D-0.0 V electrode) has been confirmed by XANES, HRTEM, and sXRD, together with spectroscopy simulations. Hence, the over-discharging reaction can be described by the following equations: $5LiCoO_2 + 3Li^+ + 3e^- \rightarrow Li_2CoO_2 + Co_3O_4 + CoO + 3Li_2O$. This means that Co²⁺ occupies 60% in the overlithiated electrode surface, which is consistent with the results of 62.8% by XPS analysis and 66.8% by XANES simulation.

## Chemical imaging analysis of D-3.0 V and D-0.0 V electrodes

X-PEEM has excellent capability in resolving surface phase separation in practical porous composite electrodes, and consequently is a powerful tool to explore battery degradation mechanisms. The spatial distribution of different phases in the D-3.0 V and D-0.0 V electrodes has been confirmed by Co chemical mapping based on fitting the Co L₃-edge PEEM image stacks, and illustrated in the red and green regions together with the Co L₃-edge XANES, as shown in Fig. 2a, d, respectively. An obvious chemical phase difference in Co can be clarified by distinguishable spectral features in the over-discharged D-0.0 V electrode in Fig. 2d, and this phenomenon does not appear in the D-3.0 V electrode in Fig. 2a. As we mentioned above, the shift of the main peak at 781 eV and the appearance of a shoulder peak located at 779 eV suggest the presence of Co²⁺, indicating Co reduction during the over-discharge process[26]. In addition, the same analysis was performed on the another two regions on the D-0.0 V electrode, as shown in Fig. S7, and the same results were obtained, showing good consistency and statistics. This means that the over-discharged LiCoO₂ electrode suffers from severe surface Co-reduction and phase heterogeneity. Spatially-resolved Raman spectroscopy was employed to further investigate the surface chemical heterogeneity and its relationship to the distribution of the conductive carbon additive during over-discharge, as seen in Fig. 2b (D-3.0 V electrode) and 2e (D-0.0 V electrode). The mapping images were constructed based on the band intensity of Co₃O₄ at 687 cm⁻¹ and the A₁g modes of LiCoO₂ at 600 cm⁻¹ [41–45]. The spatially-resolved Raman mapping images of the D-0.0 V electrode (Fig. 2e) show that more Co₃O₄-phase appears on the surface of the active particle (white dotted line region) and shows stronger intensity compared to that of the D-3.0 V electrode (Fig. 2b), indicating surface Co reduction and structure degradation after over-discharge, which is consistent with the above conclusion. Figure 2c, f display the Raman spectra extracted at the selected spots from Fig. 2b, e, respectively, and the composite electrodes produce Raman signals from both the LiCoO₂ and the conductive carbon additive. The Raman bands between 400 and 750 cm⁻¹ correspond to a mixed structure of the layered LiCoO₂ phase and the spinel Co₃O₄ phase (marked as black arrows), and the bands at 1350 and 1600 cm⁻¹ correspond to the D and G modes of the conductive carbon

additive[41,42]. Raman spectra reveal that there is a small amount of $Co_3O_4$-like phase on the surface of the active particles in the D-3.0 V electrode, as shown in Fig. 2c. Compared to other spectra in Fig. 2f, stronger intensity of the *D*-band and *G*-band in the marked *4'* spectrum in the D-0.0 V electrode suggests it is a conductive region enriched with carbon additives[46], where abundant $Co_3O_4$-like phase also forms. It can be concluded that there is a strong correlation between the formation of the $Co_3O_4$-like phase and the distribution of the conductive carbon additive. In other words, the conductive agent may accelerate the transformation of the surface structure and Co reduction of $LiCoO_2$ during the over-discharge process and the surface of the over-discharged $LiCoO_2$ electrode is characterized by phase heterogeneity.

To further explore the surface phase heterogeneity and its correlation with the conductive carbon additive and the PVDF binder, morphological and compositional mapping of the over-discharged D-0.0 V electrode was obtained by averaging the image stacks and performing on/off edge subtraction at the F, O K-edges and Co L-edge, as shown in Fig. 3. The X-PEEM elemental composite maps can be used to explore the distribution of $LiCoO_2$ crystals with different sizes, crystalline orientation, and aggregation, together with nearby environments (PVDF and carbon black). Figure S8a shows the elemental distribution mapping of the D-0.0 V electrode by X-PEEM at individual F, O, and Co edges. Further visualization of the spatial distribution of the elements was obtained by color-coded correlation maps of individual elements derived at F (green color), O (red color), and Co (blue color) edges, as shown in Fig. 3a. O and Co are mainly constrained within $LiCoO_2$ particles, which can be easily resolved, such as P1 and P2 in Fig. 3. Figures 3a and S5 illustrate that the P1 particle has a hexagonal-like shape with a smooth surface and extended sidewalls, suggesting that the particle has an exposed (001) facet, consistent with a previous report[47,48]. Furthermore, from the O and Co maps in Fig. S8, there are obvious sharp corners, edges, and lamellar shapes presented for the P2 particle, indicating that the exposed crystalline facets belong to the {010}/{100}/{110} facets, which is also displayed in the inset illustration of Fig. 3a. (Details are described in the supporting information for Fig. S8) The chemical distribution of F-containing compounds (PVDF and LiF) is shown in Fig. 3b[49]. It can be seen that PVDF is dispersed around or on the active particles. Compared with the P2 particle, a large amount of PVDF (red region in Fig. 3b) is enriched

around the P1 particle. Meanwhile, a large amount of LiF (a component of the cathode/electrolyte interphase, green region in Fig. 3b) can be found covering the surface of the active particles and displays an inhomogeneous distribution on the electrode, implying the nature of the heterogeneous degradation of the electrode during over-discharge, which is relevant to the size, the exposed crystalline planes of $LiCoO_2$ particles, and the distribution of binder/conductive agents.

Principal component analysis (PCA) is able to identify image pixels with similar spectral features, and the average of all similar pixels yields a spectrum that corresponds to a pure or mixed chemical phase[27]. The distribution of chemical phase and local electronic structure of the over-discharged D-0.0 V electrode can be mapped out by the O K-edge and Co $L_3$-edge PCA analysis, as displayed in Fig. 3c–h. Figure 3e–h are enlarged views of the P1 and P2 particles. Different color regions represent different chemical phases with the different oxidation state and local electronic structure environment. The corresponding average XANES spectra at O K-edge and Co $L_3$-edge in the different color regions were extracted from Fig. 3c–f, named as C1 to C5 phase in Fig. 3i. These XANES spectra display observable differences in chemical nature and electronic structure among these regions. The C1 phase (yellow color region) refers to the background or shadowed areas instead of active particles. In contrast, other phase regions (C2-C5) correlate well with the active particles in Fig. 3c–h, especially P1 and P2 particles, while there are large differences in the local electronic environment of O and Co between different phases, as displayed in Fig. 3i. The O pre-edge features between 529 and 532 eV reflect the hybridization of O $2p$ and Co $3d$ orbitals, and the main peaks in the range of 534-552 eV are due to transitions into O $2p$ hybridized with Co $4sp$ states. The C2 and C3 phase regions show a much lower pre-edge peak (529–532 eV) relative to the main broad features on the edge, in contrast to the C4 and C5 phases in Fig. 3i. This spectroscopic feature is a result of the lower Co-O covalence and lower Co oxidation state in C2 and C3 phases. The intensity of the $Co^{2+}$ feature (-779 eV) in C2 and C3 phases is much stronger than that in the C4 and C5 phases, which confirms the above conclusion. As we discussed with regards to Fig. 1, the CoO, $Li_2O$, $Li_2CoO_2$, and $Co_3O_4$-like phases (exclusion of $Li_xCoO_{2-y}$) are present in the over-discharged $LiCoO_2$ electrode, which is supported by O K-edge XANES simulation and Co $L_3$-edge XANES spectra. According to the O edge features in conjunction with the Co $L_3$-edge

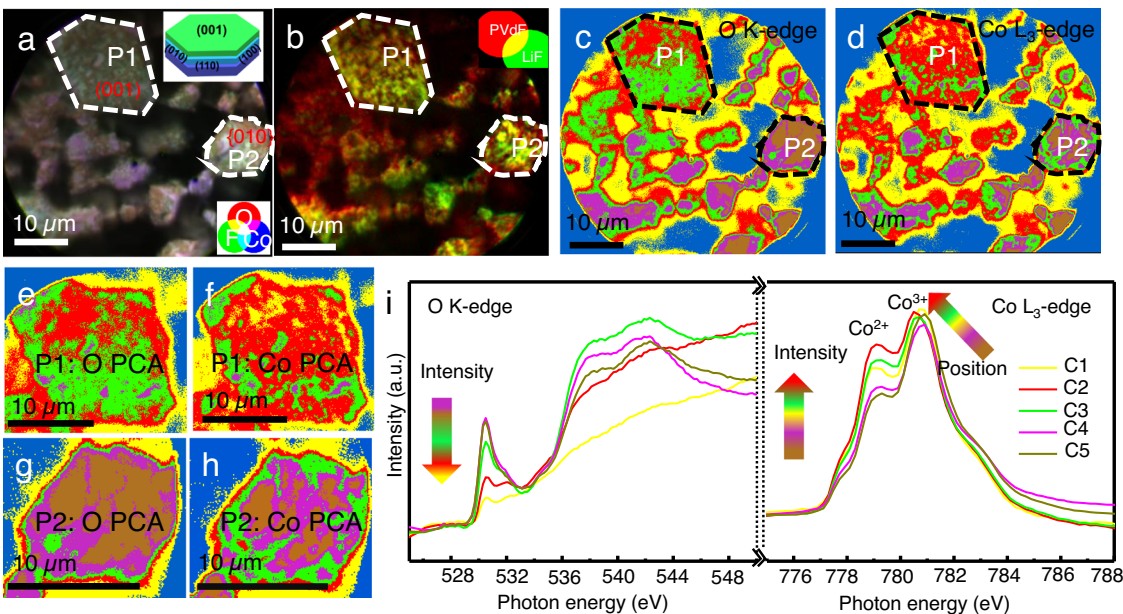

**Fig. 3 | X-PEEM chemical imaging and spatially-resolved XANES of the D-0.0 V electrode. a** X-PEEM elemental composite mapping, **b** chemical distribution of the F-containing compounds (PVDF and LiF), **c** the O K-edge PCA and **d** Co $L_3$-edge PCA

analysis of the D-0.0 V electrode; the magnified P1 (**e**, **f**) and P2 (**g**, **h**) regions in **c** and **d**; **i** O K-edge and Co $L_3$-edge XANES spectra extracted at various color regions from **c** and **d**.

features, it can be concluded that the heterogeneity nature of over-lithiation reduction follows the order of C2/C3 phase > C4/C5 phase in the over-discharged D-0.0 V electrode. The distribution of chemical phase and local electronic structure of the discharged D-3.0 V electrode are also mapped out by O and Co edges PCA for comparison, as displayed in Fig. S9. No apparent surface phase inhomogeneity and Co reduction can be observed in the discharged D-3.0 V electrode. All the spectra of the discharged D-3.0 V electrode resemble those of $LiCoO_2$, in which the O K-edge XANES is dominated by the $Co^{3+}$ feature in the pre-edge[33,50].

Additionally, such heterogeneity is also particularly evident in the spatial distribution and is closely related to the particle size and exposed crystal planes of the $LiCoO_2$ particles, as well as to the aggregation of conductive agents and binders. The O and Co PCA map out the spatial distribution of over-lithiation (i.e., $Li_{1+x}CoO_2$) in the over-discharged D-0.0 V composite electrode in Fig. 3c–h. The larger P1 particle exposes the (001) facet and is surrounded by the aggregation of PVDF and carbon, showing the predominance of C2 and C3 phases, implying a severe over-lithiation reaction accompanied by a surface structure transformation of $LiCoO_2$ to $CoO/Li_2O/Li_2CoO_2/Co_3O_4$-like phases during the over-discharge process. In contrast, the smaller P2 particle with the {010}/{100}/{110} facets exposed mainly shows C4 and C5 phases. The results of HRTEM also confirmed these conclusions. From the HRTEM images of the overdischarged particle in Fig. 1h, it can be seen that the evolution of the surface structures generally occurs on the side parallel to the (003) crystal plane, while the sides perpendicular to the (003) crystal plane (i.e., {010} facets, perpendicular to the c-axis) still retain a better-layered structure (as shown in Fig. S10). The flat (001) facet in a perfect $LiCoO_2$ particle is not active for intercalation/deintercalation of $Li^+$, while the {010}/{100}/{110} facets are believed to be the active tunnels for $Li^+$ transmission[47,51]. The surface phase heterogeneity of the over-discharge $LiCoO_2$ particles, including the (001) facet of the large P1 particle in Fig. 3, could be related to the interaction of the defective surface with the aggregation of carbon and PVDF, where electrons are allowed for fast transmission under the influence of conductive carbon. The defects on the surface of $LiCoO_2$ particles could be an intrinsic property or induced during the electrochemical formation. Such surface defects in the larger P1 particle with fast electron transmission accelerate the transformation of the surface structure and Co reduction during over-discharge. Thus, the distribution variation of $CoO/Li_2O/Li_2CoO_2/Co_3O_4$-like phases is relevant to the size and exposed crystalline planes of $LiCoO_2$ particles (the over-lithiated phases enriched in large particles with probable (001) facet), and the distribution of conductive carbon and PVDF. Very surprisingly, the spatial distribution map of the over-lithiation phase by O K-edge PCA is different from the map obtained by the Co $L_3$-edge PCA, as seen in Fig. 3c, d, respectively. Based on the O K-edge and Co $L_3$-edge XANES analysis, the C2 phase (red region) suffers from a more complete over-lithiation reaction and Co reduction than the C3 phase (green region). The amount of the over-lithiation C2 phase in the O PCA map is also lower than that in the Co PCA map. This fact may hint a different probe depth by O K-edge and Co L-edge. It can be concluded that the over-lithiation reaction and Co reduction are surface effects during discharge and its degree decreases with increasing depth. The much higher over-lithiation at surface highlights unique properties in $LiCoO_2$ surface (thermal stability and ionic transport dynamics).

## DFT calculation

To investigate the electronic structure differences between $LiCoO_2$ and a over-lithiated $Li_2CoO_2$, a comparison of their stability in different magnetic structures was carried out. For $LiCoO_2$, the singlet point energy difference between the low spin (LS) ($t_{2g}^6 e_g^0$ with a theoretical magnetic moment of 0 $\mu_b$) and high spin (HS) state ($t_{2g}^4 e_g^2$ with a theoretical magnetic moment of 4 $\mu_b$), namely, $E_{LS} - E_{HS}$, is -0.836 eV/f.u., indicating the LS configuration is more stable than the HS state of

$LiCoO_2$. In the case of $Li_2CoO_2$, the energy difference between LS ($t_{2g}^6 e_g^1$ with a theoretical magnetic moment of 1 $\mu_b$) and HS ($t_{2g}^5 e_g^2$ with a theoretical magnetic moment of 3 $\mu_b$) is 1.045 eV/f.u., demonstrating that the HS configuration is conducive to reduce the energy of the $Li_2CoO_2$ system. Such a phenomenon can be attributed to the relatively higher electron pair energies of LS-$Li_2CoO_2$ than its splitting energies. Moreover, the theoretical magnetic models of $LiCoO_2$ and over-lithiated $Li_2CoO_2$ were restricted to LS and HS, respectively, in order to further investigate their total density of states (DOS) and partial density of states (PDOS). As shown in Fig. 4a, b, the O and Co states dominate the valence bands of both $LiCoO_2$ and $Li_2CoO_2$, while the conduction bands are mainly provided by Co-O empty orbitals. Moreover, due to the strong d-d Coulomb interaction (U) in transition metal oxides, the antibonding (M−O)* band splits into one empty upper-Hubbard band (UHB) and one filled lower-Hubbard band (LHB). This splitting of Co-3d bands leads to an energy gap ($E_g$) of 2.136 and 2.215 eV for $LiCoO_2$ and $Li_2CoO_2$, respectively, which is likely to determine the lower electronic conductivity of $Li_2CoO_2$ mode than that of $LiCoO_2$ mode. This can be well explained in the above-mentioned Fig. 3, where the conductive carbon and PVDF aggregates with fast electron transmission are more likely to promote the formation of over-lithiated $Li_2CoO_2$ phase and Co reduction. Besides, the LHB of $Li_2CoO_2$ is closer to the Fermi level than that of $LiCoO_2$, corresponding to the lower electrochemical redox potential in $Li_2CoO_2$, which implies that such phase is formed under over-discharge to a lower voltage. The Co-3d and O-2p PDOS of $LiCoO_2$ and $Li_2CoO_2$ were also calculated and the details are displayed in Fig. S11. To obtain accurate electronic structures, we added theoretical calculations using the Heyd-Scuseria-Ernzerhof (HSE06) screened hybrid functional, in which an amount of exact Hartree-Fock exchange mixing parameter of 0.25 is used, and the results are shown in Fig. S12. However, as expected, the theory level impacts the band gap amplitude but not the global shape of the electronic band structures. Generally, the difference in electron structure will cause the change of electron transfer and then affect ionic diffusion dynamics. Thus, further calculation of the $Li^+$ migration barrier within the $Li_2CoO_2$ and $LiCoO_2$ frameworks was performed by using the CI-NEB method. The diffusion behavior of layered intercalation compounds has long been discussed, and two mechanisms are generally considered: oxygen dumbbell hopping (ODH) and tetrahedral site hopping (TSH)[52]. Previous studies showed that lithium diffusion in the early stage of charging (delithiation) was dominated by ODH[52,53], so we are initially based on the isolated vacancy model and adopt the ODH mechanism to study the lithium diffusion barrier. Figure 4b, e show $Li^+$ diffusion pathway within the octahedral Li sites of $LiCoO_2$ and the tetrahedral Li sites of $Li_2CoO_2$ in the same Li slab, respectively. Compared with the diffusion modes of $LiCoO_2$, the $Li^+$ diffusion barriers of $Li_2CoO_2$, is found to be very sensitive to the local environment changes induced by lithium vacancies, as seen in Fig. 4c, f. The lowest energy barriers of $Li^+$ diffusion for the above two diffusion modes are both obtained via Li dual-vacancies in the Li lattice as opposed to an isolated Li vacancy. Such Li dual-vacancies likely reduce the electrostatic repulsion between the diffused $Li^+$ and its face-sharing species[54]. The $Li_2CoO_2$ mode has a higher activation barrier (0.672 eV) than that of the $LiCoO_2$ mode (0.590 eV), which suggests the poorer $Li^+$ diffusion kinetics for $Li_2CoO_2$ mode than that of $LiCoO_2$ mode. Thus, the calculation provides a good interpretation that the over-lithiation reaction is a surface effect, and the poor ionic kinetics of $Li_2CoO_2$ further restricts the intercalation reaction of $Li^+$ during over-discharge.

## Discussion

To summarize, synchrotron-based XANES and X-PEEM, and Raman mapping combined with XANES simulations have been applied to examine the chemical heterogeneity in a commercial discharge/over-discharge $LiCoO_2$ composite electrodes. A combination of Co $L_3$-edge, O K-edge XANES, and PCA analysis offers a comprehensive

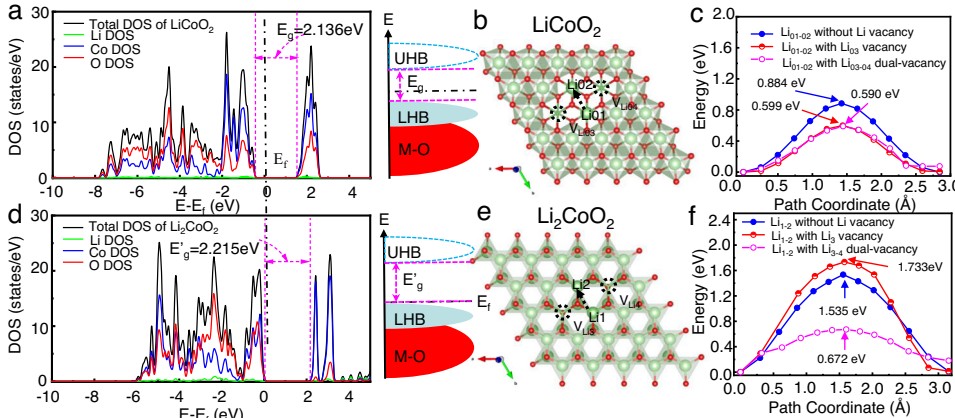

**Fig. 4 | Theoretical calculations for the electronic structures and the ionic diffusion of LiCoO₂ and Li₂CoO₂ modes.** The total density of states (DOS), the partial density of states (pDOS), and the corresponding schematic energy bands in consideration of Mott-Hubbard splitting of LiCoO₂ (**a**) and Li₂CoO₂ (**d**); The illustration of the Li⁺ diffusion pathways (**b**, **e**) and the calculated migration activation energy of the Li⁺ diffusing along with various trajectories in LiCoO₂ (**c**) and Li₂CoO₂ (**f**). $E_f$ represents the Fermi level, UHB for upper-Hubbard band and LHB for lower-Hubbard band.

visualization approach in mapping out phase heterogeneous distribution at the surface of the over-discharged LiCoO₂ in the practical composite electrode. It is found that the over-lithiation reaction is a surface effect with Co reduction and the heterogeneous degradation reaction of the surface structure to Li₂CoO₂/Co₃O₄/CoO/Li₂O-like phases (exclusion of Li₁₊ₓCoO₂₋ᵧ) during over-discharge. The surface chemical heterogeneous distribution varies with the size, depth, and exposed crystalline planes of the LiCoO₂ particle, and the distribution of binder PVDF/conductive carbon additive. Furthermore, DFT calculations provide strong evidence that lower electronic and ionic conductivity is present in the Li₂CoO₂ phase than that in the LiCoO₂ phase, which further reveals the critical effect of the aggregated distribution of binder PVDF/conductive carbon additive on the surface chemical heterogeneity of the over-lithiated LiCoO₂ particles during over-discharge. Our results emphasize that the surface heterogeneous nature of LiCoO₂ interplays with the local environments and highlight the capability of resolving spatial chemical phase heterogeneity of XANES and PEEM for studying the degradation mechanisms of LIB electrodes.

## Methods
### Electrode preparation
The industry-fabricated LiCoO₂/PE separator/Graphite pouch cells (2.7 Ah, provided by Zhuhai Guanyu Battery Co., Ltd, China) were used to study the over-discharged mechanism. The cathode electrode contained 97.2 wt.% LiCoO₂ powders with a loading about 18.85 mg cm⁻². The 2.70 Ah pouch cells were charging/discharging at 500 mA at the voltage range of 3.0–4.35 V for 10 cycles. And the cells were then further over-discharged to 0 V at 20 mA. The batteries were then disassembled under an inert environment. The cathodes were rinsed with DMC for three times and soaked in DMC for 24 h to remove the residual electrolyte, then fully dried under vacuum in a glovebox before it was transferred to PEEM in an airtight sample box for characterization.

### Electrode material characterization
X-PEEM measurements were performed at the SM beamline of Canadian Light Source (CLS). The monochromatic X-ray beam was focused using an ellipsoidal mirror to a spot of ~50 μm on the sample in PEEM at a grazing incidence angle of 16°. The sample was biased at -20 kV to promote photoemission and the base pressure of the PEEM chamber was able to maintain at ~10⁻⁹ Torr after an extended time of pumping before measurement. The incident beam intensity (Io) was simultaneously recorded by measuring the photocurrent from an Au coated Si₃N₄ window, located in the upstream vacuum line of the PEEM

main chamber, and the Io spectrum was used to normalize the acquired X-PEEM data. Image stacks for a specific FOV were acquired at the Co L-edge, and F and O K-edges with energy scan. The obtained X-PEEM data were analyzed using aXis2000 (http://unicorn.mcmaster.ca/aXis2000.html). HRTEM (JEOL JEM-2100F), EDS detector and SEM (JSM-6100LV, JEOL, Japan) were used to characterize the morphologies, element mapping and microstructures. XPS was conducted on a Thermo Scientific™ K-Alpha⁺™ spectrometer equipped with a monochromatic Al Kα X-ray source (1486.6 eV) operating at 100 W. Samples were analyzed under vacuum ($P < 10⁻⁸$ mbar) with a pass energy of 150 eV (survey scans) or 50 eV (high-resolution scans). All peaks were calibrated with the C1s peak binding energy at 284.8 eV for adventitious carbon. Synchrotron X-rays diffraction (sXRD) data was collected on the Brockhouse High Energy Wiggler beamline at the Canadian Light Source using 60.8319 keV X-rays. All data was collected in transmission mode using a Perkin Elmer area detector. For the battery experiments high-resolution diffraction data was obtained by having the high energy beam penetrate directly through the LiCoO₂ pouch cell using a long sample to detector distance of ~875 mm. All data processing was done using the GSAS-II software[55]. To confirm distribution of the Co₃O₄-like spinel phase and conductive carbon additives in the LiCoO₂ electrodes, confocal Raman mapping was conducted at the Saskatchewan Structural Sciences Center (SSSC). The Raman shifts were acquired with 80 points per line and 80 lines per image, using a scanning area of 40 μm × 40 μm. The mapping images were constructed with respect to the spinel-like phase band intensity of Co₃O₄ at 687 cm⁻¹ and the A₁g mode of LiCoO₂ at 600 cm⁻¹.

### The principle component analysis and the following cluster analysis
The principle component analysis (PCA) and the following cluster analysis (CA) of the X-PEEM spectromicroscopic data of the D-0.0 V electrode sample was performed using the PCA_GUI 1.1.1 (Stony Brook University) free software[56,57]. First, the spectral covariance of the data was calculated by multiplication of any two X-PEEM images of the same or different photon energy to generate a spectral covariance matrix. Then, the eigenvalues, eigenspectra, and eigenimages of the covariance matrix were computed. The first eigenspectrum and eigenimage with the largest eigenvalue are just average of the entire X-PEEM image stack, while the rest significant eigenspectra and eigenimages are the corrections/variations to the average until a flat eigenvalue together with noise-level eigenspectrum and eigenimage appears. These selected eigenspectra and eigenimages were used as the principle components for the following cluster analysis, but they have abstract or no physical meaning except the first principle component. For the

cluster analysis, basically a linear combination fit of the X-PEEM image stack at each pixel was performed using the selected principle components. The pixels having the same angle distance (i.e., same contribution from all the abstract principle components) were grouped and averaged to generate a color-coded X-PEEM cluster spectrum and image, which are physically and chemically meaningful. In this work, all the derived X-PEEM cluster spectra and images represent different phases or blank regions on the sample based on their spectral characteristics.

**Computational details**

**Soft X-ray Spectroscopy Calculations for $Li_{1+x}CoO_2$.** All calculations were performed for the O K-edge using WIEN2k[36], a full-potential, all-electron DFT code. The calculation details are described in the supporting information Figure S3 and S4. Because the X-ray transition in XANES leave an O 1s core-hole in the final state, an additional calculation was performed with an explicit core-hole compensated by a background charge (to model the excited photoelectron). In our previous experience, this works quite well for accurately simulating the O K-edge XANES[58–60]. The literature suggests that over-stoichiometric $Li_{1+x}CoO_{2-y}$ occurs when Li substitutes for Co ($Li_{Co}$) and has an adjacent O vacancy ($V_O$)[37]. So, we created a $2 \times 2 \times 1$ $LiCoO_2$ supercell (12 formula units of $LiCoO_2$), substituted Li at on Co site, and removed an adjacent O, to create $Li_{13}Co_{11}O_{23}$. The geometry of this structure was optimized using the PBE functional with the MSR1a mixing method[61]. After optimizing we calculated the electronic structure and O K-edge XANES using the mBJ method. The optimized calculated structure for $Li_2CoO_2$ from The Materials Project was used. The mBJ functional without a core-hole was used to simulate the O K-edge XANES. Both ferromagnetic (FM) and antiferromagnetic (AFM) structures for $Li_2CoO_2$ were tested, since $Co^{2+}$ has non-zero net spin. The AFM structure is thermodynamically favorable by a very small margin (only 6 meV/unit cell) and magnetic ordering has negligible influence on the O K-edge XANES.

**The electronic structures and ionic diffusion calculations of $LiCoO_2$ and $Li_2CoO_2$ mode.** Theoretical calculations for the lattice relaxations, electronic structures, and ionic diffusion were performed by using the CASTEP module of Materials Studio software package within the DFT framework[62]. The details are shown in the supporting information Fig. S11. Owing to considerations of precision and calculation time, supercells of $Li_{27}Co_{27}O_{54}$ (composed of 27 units of $LiCoO_2$), and over-lithiated $Li_{54}Co_{27}O_{54}$ (composed of 27 units of $Li_2CoO_2$) were used for the calculation of $Li^+$ diffusion. The minimum energy pathways of the $Li^+$ diffusion from one lattice site to the adjacent ones were investigated by the climbing image nudged elastic band (CI-NEB) method. The internal atomic positions of the initial and final structures were both optimized before the CI-NEB calculations.

## Data availability

The authors declare that all data supporting the finding of this study are available within the paper and its supplementary information files. Source data are provided with this paper.

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

## Acknowledgements

The research described in this paper was performed at the Canadian Light Source, a national research facility of the University of Saskatchewan, which is supported by the Canada Foundation for Innovation (CFI), the Natural Sciences and Engineering Research Council (NSERC), the National Research Council (NRC), the Canadian Institutes of Health Research (CIHR), the Government of Saskatchewan, and the University of Saskatchewan. We acknowledge the National Natural Science Foundation of China (Grant No. 51902072 (F.D.Y.) and 22075062 (Z.B.W.) and 21975212 (M.L.)), Heilongjiang Touyan Team (Grant No. HITTY-20190033 (Z.B.W.)), Heilongjiang Province "hundred million"-project science and technology major special projects (2019ZX09A02; Z.B.W.), State Key Laboratory of Urban Water Resource and Environment (Harbin Institute of Technology No. 2020DX11(Z.B.W.)) and the Fundamental Research Funds for the Central Universities (Grant No. FRFCU5710051922; Z.B.W.), China postdoctoral science foundation (Grant No. 2021M702256 (G. S.)). We thank Instrument Analysis Center of Shenzhen University for the assistance with HRTEM and SEM analyses. Thanks to Dr. Graham King for his help in the XRD measurement of the battery over discharge experiment and data analysis.

## Author contributions

G.S. and F.Y. conceived the idea, designed the experiments, and wrote the manuscript. G.S., J.Z., J.W., and Z.W. carried out the electrochemical test, characterization, and analyzed the results with the help of M.L., Q.Z., Y.J., Y.M. J.M., J.M., and F.Y. conducted the DFT calculations.

## Competing interests

The authors declare no competing interests.

## Additional information

**Correspondence and requests** for materials should be addressed to Jian Wang, Jigang Zhou or Zhenbo Wang.

