## [Peer Review File · Nature Communications]

Surface chemical heterogeneous distribution in over-lithiated $\text{Li}_{1+x}\text{CoO}_2$ electrodesEditorial Note: Parts of this Peer Review File have been redacted as indicated to remove third-party material where no permission to publish could be obtained.

REVIEWER COMMENTS

Reviewer #1 (Remarks to the Author):

Please find the attachment.

The authors have performed a series of characterisation techniques including XANES, X-PEEM and Raman to study the phase distribution of over-lithiated LiCoO₂ electrode within a commercial cell, aiming to elucidate the over-lithiation mechanism of this important cathode material. Whilst the reviewer agrees on the importance of this research topic to the battery community, the presented data in the current form however lacks a rigorous analysis and interpretation, with some of the key conclusions remaining to be substantiated.

1) According to reference 32 and 33, the model compound Li_{1+x}CoO_{2-y} only contains Co³⁺. However, Co²⁺ was observed in the Co L-edge spectrum (Fig. 1b). Therefore, is it not to be expected that the calculated O K-edge (based on a purely Co³⁺ containing species) will differ from the experimental data? Also, comparing experimental data against a library of reference spectra solely derived from simulation is inappropriate. Note that the experimental O K-edge spectrum of CoO (in literature) appears notably different from the calculated spectrum in the manuscript (see Fig. r1). In addition, the authors later stated that a linear superposition of LiCoO₂, Li₂CoO₂ and Co₃O₄ may provide the best fit to the experimental data. But where are this fit and the corresponding quantification? Whilst formation of Li₂CoO₂ and Co₃O₄ upon overlithiation of LiCoO₂ does make sense, the current analysis appears too qualitative and lacks convincing crystallographic support via XRD or electron diffraction, which in the reviewer's opinion, is prerequisite for the later discussion of the phase distribution.

[REDACTED]

Fig. r1. a-b) experimental O K-edge spectra for Co₃O₄ and CoO, adapted from Tongfei Shi et al 2009 J. Phys.: Conf. Ser. 193 012099. c) taken from the manuscript.

2) According to the reviewer's understanding, the number of distinct phases (c1-c5) and their spatial distribution was derived from the PCA analysis of a library of XANES spectra taken from each pixel (of the X-PEEM mapping data). But the PCA analytical procedure was not provided in the manuscript, nor was it elaborated in reference 22. The resulting ambiguity renders the data interpretation a bit difficult to digest. How was the number of distinct phases (i.e. 5) determined? If this value was deduced based on the PCA output (i.e. eigen values), the authors need to show these values so that it can be justified. In addition to this, what do the spectra in Fig. 3i represent — are they reconstructed spectra using the PCA output or merely experimental data? And following this question, what does the colour in Fig 3c-h mean – does that reflect the counting sale in the experiment or results from the PCA analysis? Concerning the physical meaning of these C1-C5 phase, while the C1 phase was attributed to the background, how do the C2-C5 spectra relate to the as-determined Li₂CoO₂/LiCoO₂/Co₃O₄-like phase?

3) If the authors ought to deliver a robust and compelling characterisation of spatial distribution in a multi-phase system, not only a quantitative and reliable examination of the crystallite shape anisotropy is essential, but also an evaluation from multiple particles to achieve better statistics is desirable. The reviewer understands that it might be challenging to spot several particles (Fig. 3a-b) with sufficiently large sizes to determine the crystal orientation in Fig. 3a, but some experimental effort (e.g. via ED) to verify that the electrode (Fig. S1) is indeed dominated by crystals showing $\{110\}$ and $\{010\}$ facets is necessary. After all, it is difficult to find a hexagonal plate-like particle in Fig. S1, how can the authors convince the readers that the P1 particle has such a morphology?

Reviewer #2 (Remarks to the Author):

The paper describes the phases present at the surface of an over-discharged LiCoO₂ battery electrode. It employs X-ray spectroscopies and Raman in close conjunction, with very good correlation between the techniques to provide a better understanding of the phase behaviour than would be gained by using them individually. It is not easy to get that consistency when using these imaging techniques so the authors should be pleased to have been able to bring things together so successfully. The experimental work is backed up by modelling to allow explanation of some of the spectral features. Overall I found this paper to be thorough and well-argued, and an important contribution to the understanding of this system under this type of abuse conditions. It seems suitable for publication as written.

Reviewer #3 (Remarks to the Author):

This is an interesting paper about surface degradation at overlithiation. It is overall good work and will be important in this field. However, some of the key points are not clearly supported. I think it could be published in NCom, but substantial modifications and more support by data will be necessary.

1. It is both mysterious and interesting that the authors claim overlithiation is harmful to electrodes, which somehow contradicts the observations of two recent publications Nature volume 585, pages 63–67 (2020) and <https://doi.org/10.1038/s41563-022-01242-0>. It will be helpful for people in this field if the authors can discuss in-depth the fundamental difference between their observation and the reported work.
2. Some of the figures are poorly polished, such as Figure 1(a)-(c). They have very low resolution and small font sizes. Same problem applies in Figure 2(c), (f), Figure 4 (a)(c)(d)(f).
3. The comparisons in Figure 1(c) are great. However, the authors may also want to compare the computed XANES of potential formation of overlithiated rocksalt phase, a model that can be referred to in the two papers listed in my comment 1. CoCO₃ is also a relevant phase that has Co reduction and can be frequently seen at the surface.
4. Since the authors are trying to compare measured XANES with computed XANES. It is very critical to benchmark the reliability of the functions they use. As shown in Figure S2, the peak position is very sensitive to functionals. The authors should compare their computed XANES with experimental measured XANES for at least some of the well-known materials such as CoO₂, Co₃O₄, Li₂O, CoO, etc.
5. It shows in Figure S3 that computed XANES for Li₁₃Co₁₁O₂₃ is very sensitive to ionic ordering. It is probably also sensitive to stoichiometry. In order to justify Li_{1+x}CoO_{2-y} can never show XANES that is similar to what has been observed at 0.0 V. More systematic evidence should be provided.
6. While the authors have much better evidence about Co₃O₄ existence, I have concerned that the evidence that Li₂CoO₂ is also formed is very weak. It should also be mentioned that the forming of spinel M₃O₄-like structure is very common in both LiCoO₂ and LiNiO₂, which can come from densification and Li loss at the surface. Surface reduction is very common in cathode materials, particularly after cycling. Thus the existence of Co₃O₄ at the surface can be totally irrelevant to overlithiation. The authors really want to have more convincing evidence to support that.
6. Some DFT details are missing from Figure 4. Are these computations done by GGA? GGA is well known to have the wrong prediction of electronic structure in Co-based layered materials. Hybrid functional should be a better way to quantify the bandgap.
7. In addition to the comment above, it is not convincing to claim that 0.079 eV of bandgap increase will lead to worse electronic conductivity. Electronic conductivity is related to both carrier concentration and mobility, the bandgap only controls the carrier concentration, while it is possible

that Li_2CoO_2 will have better electronic mobility.

8. It appears to me that the performance degradation of LiCoO_2 is irrelevant to the bulk property of formed Li_2CoO_2 and Co_3O_4 . Or at least, is not well supported by the data. In addition to the almost neglected difference in electronic conductivity between LiCoO_2 and Li_2CoO_2 and the mysterious diffusion barrier as mentioned above. Co_3O_4 spinel structure with a bit of off-stoichiometry will be a perfect framework for Li diffusion, which should be similar to the high rate spinel-based Li-Mn-O spinel cathodes. I am not sure if there is a reason that the authors did not show any DFT results in the Co_3O_4 phase.

9. The way the activation barrier is calculated is somehow inconsistent with the literature. As indicated by Fig. 4b, the $\text{LiO}_1 \rightarrow \text{LiO}_2$ diffusion follows the oxygen dumbbell hop, which is demonstrated in previous papers to be unfavorable and unlikely to happen in LiCoO_2 . Somehow the authors demonstrated a much lower activation barrier than the literature, even without the appearance of divacancy. The authors should really check if their calculations are wrong, or at least wrote some comments to compare their results with existing literature such as *Journal of Power Sources* 97-98 (2001) 529-531.

Reviewer #1:

Comments: The authors have performed a series of characterisation techniques including XANES, X-PEEM and Raman to study the phase distribution of over-lithiated LiCoO₂ electrode within a commercial cell, aiming to elucidate the over-lithiation mechanism of this important cathode material. Whilst the reviewer agrees on the importance of this research topic to the battery community, the presented data in the current form however lacks a rigorous analysis and interpretation, with some of the key conclusions remaining to be substantiated.

Respond: Thank you for your constructive comments on this work. We have added some new pieces of evidence and modified the manuscript according to your suggestions. We believe that after our improved interpretations, modifications, additional work and analysis, our conclusions can be scientifically substantiated. **The following is our point-by-point reply to your questions and comments.**

Question 1: According to reference 32 and 33, the model compound Li_{1+x}CoO_{2-y} only contains Co³⁺. However, Co²⁺ was observed in the Co L-edge spectrum (Fig. 1b). Therefore, is it not to be expected that the calculated O K-edge (based on a purely Co³⁺ containing species) will differ from the experimental data?

Answer 1: Thanks for your comments. In this manuscript, the O K-edge XANES spectrum of the over-discharged (D-0.0V) electrode shows some difference compared to that of the D-3.0V electrode and the LiCoO₂ reference, **which displays the reduced π^* (~530 eV) intensity relative to that of the σ^* (536-542 eV).** The O pre-edge features between 529 and 532 eV reflect the hybridization of O 2p and Co 3d orbitals, and the main peaks in the range of 534-552 eV are due to transitions into O 2p hybridized with Co 4sp states. Therefore, the valence state of Co plays a key role in the local electronic structure of O. **We agree with the reviewer that both Co²⁺ and Co³⁺ were observed in the Co L-edge spectrum (Fig. 1b) of the over-discharged (D-0.0V) electrode.** For the influence of Co²⁺ and Co³⁺ species on O K-edge XAS, we compared the O K-edge XAS of the D-0.0V electrode with that of LiCoO₂, Co₃O₄ and CoO (including both the experimental and calculated data). In addition, in order to better fit

the experimentally obtained O K-edge XAS of the D-0.0V electrode, we also provide the experimental and calculated O K-edge XAS of Li_2O , Li_2CO_3 , and CoO_2 , as seen in Fig. R1. **Previous reports, such as reference 32 and 33, provided the $\text{Li}_{1+x}\text{CoO}_{2-y}$ model to explain the phenomenon of overlithiation of LiCoO_2 . Here, we mainly use this model to rule out the possibility of forming oxygen vacancies due to overdischarge.** If oxygen vacancies are generated, they will lead to severe distortion of the oxygen spectrum around the vacancies. We have related statements in the supporting information that the oxygen vacancies will lead to a small peak which appears before 529 eV in the O K-edge XAS, as shown in Fig.R1 (As the arrow points.). It is weakly related to the valence state of $\text{Co}^{2+/3+}$, as it does not appear in the oxygen spectrum of either calculated or actual CoO and Co_3O_4 (containing Co^{2+} ions). The calculated $\text{Li}_{1+x}\text{CoO}_{2-y}$ spectrum is different from the experimental one, therefore we excluded the oxygen vacancy model. **Necessary explanations and modifications have been made in revised manuscript, which are shown in Fig. 1 and Fig. S2.**

Fig. R1 the experimental and calculated O K-edge spectra for Li_2CO_3 , $\text{Li}_{1+x}\text{CoO}_{2-y}$, CoO , Li_2O , Co_3O_4 , LiCoO_2 and D-3.0V, D-0.0V

Question 2: Also, comparing experimental data against a library of reference spectra solely derived from simulation is inappropriate. Note that the experimental O K-edge

spectrum of CoO (in literature) appears notably different from the calculated spectrum in the manuscript (see Fig. r1).

Answer 2: Thank you for your constructive suggests. We have added the experimental O K-edge XANES of CoO, Li₂O, Co₃O₄, LiCoO₂ and Li₂CO₃ in the revised manuscript to compare with that of experimental discharge/over-discharged electrode, as seen in Fig. 1 (revised manuscript), Fig. R1 and Fig. S2 (supporting information). In addition, since Li₂CoO₂ and Li_{1+x}CoO_{2-y} are not available in the pure format, we can only provide the calculated O K-edge spectrum as a reference. Here, we also calculated the O spectrum of CoO, Li₂O, Co₃O₄, LiCoO₂ and Li₂CO₃ materials for comparison with the experimental spectrum.

Since the reference material may have impurities, defects and other shortcomings, there are certain differences between the experimental and calculated O spectra, but it needs to be emphasized that the calculated spectra well replicated all the main features of the experimental spectra from the reference compounds. **In the revised manuscript, we provide the experimental O K-edge spectrum of CoO, which also appears notably different from that in Figure R2a and R2b (the reviewer provided, *Journal of Physics: Conference Series* 2009, 193, 12099.), but it is similar to the O K-edge spectrum of CoO in Figure R2c and 2d (*Journal of Physics: Condensed Matter* 2008, 20, 255236; *Appl. Phys.* 2007, 102, 14908.). In fact, a broader survey of the literature (see all references cited under CoO at the XAS/EELS database for O 1s spectra: <https://anorg.chem.uu.nl/xaseels/XASEELS%20O1s.html>), there is considerable variation in the intensity of the low-energy features. In our experience, it is extremely difficult to obtain reliable spectra of MnO, FeO, and CoO (especially for commercial powders) as they are all prone to surface oxidation – and even minor oxidation tends to introduce a lot of sharp pre-edge features. In the present case, it is likely that surface Co₃O₄ produces the low-energy shoulder near 530 eV. This hypothesis is supported by a transmission EELS O 1s spectrum performed on crystalline CoO (see *Ultramicroscopy* 2003, 96, 469-480) which almost completely lacks this low-energy feature. Consequently, we believe the shape of our calculated**

Editorial Note: The authors wish to indicate that the reference citations for panel c & d were accidentally inverted in the below legend, and the correct information is as follows: J. Appl. Phys. 2007, 102, 14908(c); Journal of Physics: Condensed Matter 2008, 20, 255236 (d). Panel (c) Reprinted from the reference citation above, with permission of AIP Publishing. Panel (d) reprinted with permission of IOP Publishing, Ltd, reference citation above, permission conveyed through Copyright Clearance Center, Inc.

CoO spectrum is reasonably accurate and any discrepancy in pre-edge features is more likely due to sample contamination than an error in the calculation.

[REDACTED]

Fig. R2 Experimental O K-edge spectra for Co₃O₄ and CoO, adapted from Journal of physics. Conference series 2009, 193, 12099 (a, b); Journal of Physics: Condensed Matter 2008, 20, 255236 (c); Appl. Phys. 2007, 102, 14908 (d).

Question 3: In addition, the authors later stated that a linear superposition of LiCoO₂, Li₂CoO₂ and Co₃O₄ may provide the best fit to the experimental data. But where are this fit and the corresponding quantification?

Answer 3: Thanks for your comment. We have performed the fitting with better reference spectra. **The linear combination fit (LCF) of D-0.0V and D-3.0V samples by experimental CoO, Li₂O, Co₃O₄ and LiCoO₂, and calculated Li₂CoO₂ are displayed in Fig. R3. The LCF of the experimental CoO, Li₂O, Co₃O₄, LiCoO₂ and calculated Li₂CoO₂ components to the O K-edge XANES spectra of D-3.0V and D-0.0V samples, allowing only energy shifting and intensity scaling of each component while keeping the overall spectral shape, yield the best fitting correspondence.** The fit yielded LiCoO₂, Co₃O₄ and CoO contributions of 77.3, 14.9 and 7.8% to the O K-edge XANES of D-3.0V, respectively. The fit yielded LiCoO₂, Co₃O₄, CoO, Li₂O and calculated Li₂CoO₂ contributions of 8.1, 22.9, 21.9, 29.3 and 17.7% to the O K-edge XANES of D-0.0V, respectively. **Necessary explanations and modifications have been made in the revised manuscript, which are shown in Fig. 1c and 1d.**

Figure R3 a. Linear combination fit of O K-edge XANES of D-3.0V and D-0.0V samples by experimental CoO, Li₂O, Co₃O₄ and LiCoO₂, and calculated Li₂CoO₂; **b.** Fit component comparison of D-3.0V and D-0.0V samples.

Question 4: Whilst formation of Li₂CoO₂ and Co₃O₄ upon overlithiation of LiCoO₂ does make sense, the current analysis appears too qualitative and lacks convincing crystallographic support via XRD or electron diffraction, which in the reviewer's opinion, is prerequisite for the later discussion of the phase distribution.

Answer 4: Thanks for your constructive comments. According to your suggestion above (**Question 3**), we made a linear combination fitting of O K-edge XANES of D-3.0V and D-0.0V electrodes. The results indicate the surface structure transformation of the LiCoO₂ electrode during the overdischarge process (structure transition from LiCoO₂+Li⁺ to Co₃O₄, CoO, Li₂O and Li₂CoO₂). **Based on the LCF analysis, the quantitative results on the contribution of each component were obtained, as we discussed in Answer 3.** In addition, the XPS, HRTEM and in-situ sXRD analyses were also performed to investigate the surface chemical/structure evolution of LiCoO₂ electrodes during overdischarge, and the results are shown in Figures R4, R5 and R6 (Additional discussions are added in the revised manuscript and supporting information). The XPS spectra of D-3.0V and D-0.0V electrodes were used to detect the changes in chemical compositions in Figure R4. **Compared with that of the D-3.0V electrode (29.2% for Co²⁺), a large amount of Co²⁺ (62.8%) was found on the**

surface of the D-0.0V electrode, suggesting the surface Co reduction during overdischarge (Reduction to low-valent cobalt oxides after overdischarge, such as $\text{Li}_2\text{CoO}_2/\text{CoO}/\text{Co}_3\text{O}_4$). These results are highly consistent with the conclusions of the XANES spectra. The HRTEM and corresponding FT/IFT images in Figure R5 clearly indicated the existence of the cubic $\text{CoO}/\text{Co}_3\text{O}_4/\text{Li}_2\text{O}$ phase. Further analysis was performed on the 4 regions in Figure R5a: R1 represents the outer layered structure; R2 represents the inner layered structure; R3 represents the Co_3O_4 region; R4 represents the CoO region. The lattice fringes of a representative layer structure with a d-spacing of 4.90 Å in R1 and 4.76 Å in R2 could be assigned to the (003) plane of layered LiCoO_2 . **The $d_{(003)}$ -spacing in the outer layered structure R1 region is greater than that in the R2 region, which may be due to the expansion of the interlayer spacing caused by the intercalation of an over-stoichiometric Li^+ into the Li layer during the overlithiation process (the layered LiCoO_2 transformed to the layered Li_2CoO_2).** The lattice fringe with a d-spacing of 2.86 Å in R3 region could be assigned to the (220) plane of spinel Co_3O_4 (JCPDS file nos. 43–1003, $a = 8.084$ Å). It was worth noting that the R4 region exhibited a clear lattice fringe with a d-spacing of 2.13 Å, corresponding to the (200) plane of CoO (JCPDS file nos. 78–0431, $a = 4.2667$ Å). (**Nat. Commun. 2015, 6. Doi:10.1038/ncomms9106**) In addition, for the surface of the D-0.0V electrode a thin coating with thickness 2-4 nm was bound to the surface of the LiCoO_2 structure and distributed in regions in Figure R5b. It was confirmed from the enlarged area and the FT image that the outer layer in Figure R5b is Li_2O phase, in which a d-spacing of 0.263 nm corresponds to the (111) plane of Li_2O . (**Energy Storage Materials 2022, 51, 306-316.**)

Furthermore, in-situ synchrotron XRD of a $\text{LiCoO}_2/\text{graphite}$ pouch cell was also used to investigate the structural evolution of the LiCoO_2 electrode during overdischarge from 3.0 to 0.0 V at 20 mA g^{-1} , and the result is shown in Figure R6. All the diffraction peaks have no significant change (the position of the main peak) and no impurity peaks appear (such as $\text{CoO}/\text{Co}_3\text{O}_4/\text{Li}_2\text{O}$ phase), only a slight difference in intensity, which indicates that overlithiation is only a surface phenomenon and has little effect on the bulk structure. However, a small enhancement of (006) peak intensity of

the LiCoO_2 phase can be observed during the overdischarge process, while the (012) peak intensity has no intensity change (insets of **Figures R6a and R6b**). All synchrotron XRD data were collected under the same conditions, so the intensity change in the diffraction peak could only be due to introduction of atom(s) in the unit cell, which resulted in a change in the *structure factor*. **So, the (006) peak increase may indicate excessive intercalation of Li^+ in the Li layer, which in turn produces the Li_2CoO_2 structure, while the influence on other directions is small.**

Figure R4 Core-level XPS spectra of Co 2p of D-3.0V and D-0.0V electrodes.

Figure R5 HRTEM and the corresponding FT/IFT images of D-0.0V electrode.

Figure R6 In-situ synchrotron XRD of the overdischarged LiCoO_2 /graphite pouch cell (Discharge from 3.0V to 0.0V at 20 mA g^{-1}).

Question 5: According to the reviewer's understanding, the number of distinct phases (c1-c5) and their spatial distribution was derived from the PCA analysis of a library of XANES spectra taken from each pixel (of the X-PEEM mapping data). But the PCA

analytical procedure was not provided in the manuscript, nor was it elaborated in reference 22. The resulting ambiguity renders the data interpretation a bit difficult to digest. How was the number of distinct phases (i.e. 5) determined? If this value was deduced based on the PCA output (i.e. eigen values), the authors need to show these values so that it can be justified. In addition to this, what do the spectra in Fig. 3i represent — are they reconstructed spectra using the PCA output or merely experimental data? And following this question, what does the colour in Fig 3c-h mean — does that reflect the counting rate in the experiment or results from the PCA analysis? Concerning the physical meaning of these C1-C5 phase, while the C1 phase was attributed to the background, how do the C2-C5 spectra relate to the as-determined $\text{Li}_2\text{CoO}_2/\text{LiCoO}_2/\text{Co}_3\text{O}_4$ -like phase?

Answer 5: Thanks for your comments. We have added the details of the PCA analytical procedure to the revised manuscript in the **EXPERIMENTAL SECTION**, as follow:

The principle component analysis and the following cluster analysis

The principle component analysis (PCA) and the following cluster analysis (CA) of the X-PEEM spectromicroscopic data of the D-0.0V electrode sample was performed using the PCA_GUI 1.1.1 (Stony Brook University) free software. (Ultramicroscopy 100 (2004) 35–57; Journal of Electron Spectroscopy and Related Phenomena 144–147 (2005) 1137–1143) First, the spectral covariance of the data was calculated by multiplication of any two X-PEEM images of the same or different photon energy to generate a spectral covariance matrix. Then, the eigenvalues, eigenspectra, and eigenimages of the covariance matrix were computed. The first eigenspectrum and eigenimage with the largest eigenvalue are just average of the entire X-PEEM image stack, while the rest significant eigenspectra and eigenimages are the corrections/variations to the average until a flat eigenvalue together with noise-level eigenspectrum and eigenimage appears. These selected eigenspectra and eigenimages were used as the principle components for the following cluster analysis, but they have abstract or no physical meaning except the first principle component. For the cluster analysis, basically a linear combination fit of the X-PEEM image stack at each pixel

was performed using the selected principle components. The pixels having the same angle distance (i.e., same contribution from all the abstract principle components) were grouped and averaged to generate a color-coded X-PEEM cluster spectrum and image, which are physically and chemically meaningful. In this work, all the derived X-PEEM cluster spectra and images represent different phases or blank regions on the sample based on their spectral characteristics.

Regarding the correlation of the C2-C5 spectra with the identified LiCoO₂/Li₂CoO₂/Co₃O₄/CoO/Li₂O-like phases, we have given the corresponding description in Figure 1 of the revised manuscript. Firstly, for the O K-edge XANES of LiCoO₂ phase in Figure 1, there is a stronger π^* (~530 eV) intensity relative to that of the σ^* (536-542 eV), accompanied by a lower Co L-edge XANES shoulder (pointing to Co²⁺ at 779 eV). **According to the O K-edge XANES spectra of C2-C5 in Figure R9i, it can be seen that the reduced π^* (~530 eV) intensity relative to that of the σ^* (536-542 eV), and a significant shoulder (Co²⁺) is observed on the Co L₃-edge XANES spectra, which is significantly different from the characteristics of LiCoO₂ phase. Based on the analysis of O K-edge and Co L₃-edge XANES spectra along with O K-edge XANES simulations, the existence of LiCoO₂, CoO, Li₂O, Li₂CoO₂ and Co₃O₄ phases can be inferred.** What's more, the C2 and C3 phase regions show a much lower pre-edge peak (529-532 eV) relative to the main broad features on the edge, in contrast to the C4 and C5 phases in **Figure R9i**. This spectroscopic feature is a result of the lower Co-O covalence and lower Co oxidation state in C2 and C3 phases. The intensity of the Co²⁺ feature (~779 eV) in C2 and C3 phases is much stronger than that in the C4 and C5 phases, which confirms the above conclusion. As we discussed with regards to **Figure 1**, the CoO, Li₂O, Li₂CoO₂ and Co₃O₄-like phases (exclusion of Li_xCoO_{2-y}) are present in the over-discharged LiCoO₂ electrode, which is supported by O K-edge XANES simulation and Co L₃-edge XANES spectra. According to the O edge features in conjunction with the Co L₃-edge features, it can be concluded that the heterogeneity nature of over-lithiation reduction follows the order of C2/C3 phase > C4/C5 phase in the over-discharged D-0.0V electrode.

Figure R7 O PEEM

Figure R8 Co PEEM

Figure R9 (a) X-PEEM elemental composite mapping, (b) the chemical distribution of F-containing compounds (PVDF and LiF), (c) the O edge PCA and (d) Co edge PCA analysis of the D-0.0V electrode; the magnified P1 (e and f) and P2 (g and h) regions in (c) and (d); O K-edge and Co L₃-edge XANES spectra extracted at various color regions from (c) and (d).

Question 6: If the authors ought to deliver a robust and compelling characterisation of spatial distribution in a multi-phase system, not only a quantitative and reliable examination of the crystallite shape anisotropy is essential, but also an evaluation from multiple particles to achieve better statistics is desirable. The reviewer understands that it might be challenging to spot several particles (Fig. 3a-b) with sufficiently large sizes to determine the crystal orientation in Fig. 3a, but some experimental effort (e.g. via ED) to verify that the electrode (Fig. S1) is indeed dominated by crystals showing {110} and {010} facets is necessary. After all, it is difficult to find a hexagonal plate-like particle in Fig. S1, how can the authors convince the readers that the P1 particle has such a morphology?

Answer 6: Thanks for your comments. Additional XRD, SEM and HRTEM data were further probed and analyzed in order to better understand the exposed crystal planes of the pristine LiCoO₂ particles and the surface structure information of the overdischarge electrode. The sXRD data in Figure R6 show that the peak intensity of (003) is the strongest, indicating that (003) is the dominant crystal plane for particle growth (the grain has the priority to growth along the c-axis, [001] direction). In addition, the lateral planes of the (003) facet would be wrapped by the {010}/{100}/{110} facets, so the (003) facet and the {010}/{100}/{110} facets would be preferentially exposed on the single crystal grains of LiCoO₂. The SEM images of the pristine LiCoO₂ particles (Figure R10a) and D-0.0V electrode (Figure R10b) also reveal these large single crystal primary particles with different exposed facets. **The SEM images delineate step edges, as expected for layered LiCoO₂ grains (Figure R10a and R10b), and a similar phenomenon is observed on P2 in Figure 3a and 3b. The particles with a smooth surface and extended sidewalls suggest that the particle has an exposed (001) facet (as marked regions in red dotted line in SEM images). This conclusion has also been confirmed in the previous literature (Nat. Nanotechnol. 2010, 5, 749-754; ACS Appl. Mater. Inter. 2016, 8, 2723-2731).** At the same time, multiple particles were explored on the D-0.0V electrode in Figure 10b to achieve better statistical data. Besides, the obvious step edges and lamellar shapes, marked in yellow arrows in Figure R10, capped by a (001) facet, parallel to the c-axis, indicating that the exposed crystalline facets belong to the {010}/{100}/{110} facets.

Furthermore, in the manuscript, based on the O and Co PCA mapping, it can be concluded that the particle with an exposed (001) facet exhibit severe over-lithiation reactions accompanied by a surface structure transformation of LiCoO₂ to Li₂CoO₂/CoO/Li₂O/Co₃O₄-like phases during the over-discharge process instead of exposing one side of the {010}/{100}/{110} facets. The results of HRTEM data analysis also confirmed these conclusions. **From the HRTEM images of the overdischarge particle in Figure R5a, it can be seen that the evolution of the surface structure generally occurs on the side parallel to the (003) crystal plane, while the side perpendicular to the (003) crystal plane ({010} facets, perpendicular**

to the *c*-axis) its surface still retains a better-layered structure (as shown in Figure R10c).

Therefore, our conclusions are derived from the analysis results of multiple particles and the mutual support of a large amount of data, and provide strong and compelling spatial distribution characteristics in multiphase systems. Necessary interpretations, modifications, additional work and analysis have been made in the revised manuscript and supporting information.

Figure R10 The SEM images of pristine LiCoO₂ particles (a) and D-0.0V electrode (b); (c) HRTEM image of D-0.0V electrode.

Reviewer #2:

Comments: The paper describes the phases present at the surface of an over-discharged LiCoO₂ battery electrode. It employs X-ray spectroscopies and Raman in close conjunction, with very good correlation between the techniques to provide a better understanding of the phase behaviour than would be gained by using them individually. It is not easy to get that consistency when using these imaging techniques so the authors should be pleased to have been able to bring things together so successfully. The experimental work is backed up by modelling to allow explanation of some of the spectral features. Overall I found this paper to be thorough and well-argued, and an important contribution to the understanding of this system under this type of abuse conditions. It seems suitable for publication as written.

Respond: Thanks very much for your comments and your recognition of our work.

Reviewer #3:

Comments: This is an interesting paper about surface degradation at overlithiation. It is overall good work and will be important in this field. However, some of the key points are not clearly supported. I think it could be published in NCom, but substantial modifications and more support by data will be necessary.

Respond: We appreciate the referee's opinion of our work and have made every effort to address their concerns and clarify our arguments. We hope our revised manuscript will meet their approval. After our improved interpretations, modifications, additional work and analysis, we believe that our conclusions can be scientifically substantiated.

The following is our point-by-point reply to your questions and comments.

Question 1: It is both mysterious and interesting that the authors claim overlithiation is harmful to electrodes, which somehow contradicts the observations of two recent publications Nature volume 585, pages63–67 (2020) and <https://doi.org/10.1038/s41563-022-01242-0>. It will be helpful for people in this field if the authors can discuss in-depth the fundamental difference between their observation and the reported work.

Answer 1: Thanks for your comments. The referee raises an important point. We have added a paragraph to our revised manuscript in the **Introduction** section discussing the different processes of overlithiation in different cathode materials.

Introduction

However, recent results reveal that the impact of over-discharge is highly dependent on the local structure of the cathode: if the cathode material can safely and reversibly accommodate excess lithium, over-discharge can even be beneficial for battery performance. Over-lithiating the cathode before the first cycle offers the promise of retaining full battery capacity if the cathode material was specially prepared to safely accommodate the excess lithium (**Adv. Energy Mater.** 2021, 11, 2101565). Improved performance by over-discharging was also demonstrated in $\text{Li}_3\text{V}_2\text{O}_5$ and

Li₃Nb₂O₅ cathodes, which adopt a crystal structure that has plenty of vacancies to accommodate the excess lithium (**Nature** 2020, 585, 63-67; **Nat. Mater.** 2022, 21, 795-803). On the other hand, over-discharging LiMn₂O₄ was shown to significantly degrade performance, as the M₃O₄ spinel structure does not easily accommodate excess metal ions (**J. Solid State Electr.** 2017, 21, 3269-3279). In particular, the battery performance of Li- and Mg-doped LiCoO₂ was shown to be resilient to over-discharge compared to native LiCoO₂ (**Materials Today Energy** 2022, 27, 101040), however the location of the excess Li in LiCoO₂ was not determined. These findings make it evident that detailed probes of local structure are needed to elucidate the process of over-discharge.

Question 2: Some of the figures are poorly polished, such as Figure 1(a)-(c). They have very low resolution and small font sizes. Same problem applies in Figure 2(c), (f), Figure 4 (a)(c)(d)(f).

Answer 2: Thanks for your comment. All the figures have been overhauled to a high quality standard to meet the requirements.

Question 3: The comparisons in Figure 1(c) are great. However, the authors may also want to compare the computed XANES of potential formation of overlithiated rocksalt phase, a model that can be referred to in the two papers listed in my comment 1. CoCO₃ is also a relevant phase that has Co reduction and can be frequently seen at the surface.

Answer 3: Thanks for your comments. The referee raises an interesting point. **Certainly, the referenced findings of overlithiated rocksalts in vanadium and niobium oxides could be relevant. However, after a careful consideration, we believe that the overlithiated rocksalt cobalt oxide is not appropriate for this work.** Our reasoning is as follows.

Stoichiometric NbO forms a rocksalt phase with “ordered vacancies” (*see Physical Review B* 1993, 48 16986, for example), in which one of the four primitive face-centered sites is empty for both Nb and O sublattices. Furthermore, stoichiometric Nb₂O₅ forms a monoclinic structure that contains structures that approximate a cube with Nb at the corners and O on the edges. In this context, there are empty sites available

to incorporate Li to form the rocksalt structure: there is a reasonable pathway to a lithiated rocksalt by adding both Li and O to NbO, or by adding Li to Nb₂O₅. (And, presumably, the overlithiated rocksalt is only favorable in Li-rich conditions.)

Although stoichiometric VO is a normal rocksalt, stoichiometric V₂O₅ forms a similar monoclinic structure as Nb₂O₅ with the same V-O cube substructure. Given the chemical similarity between Nb and V it seems reasonable that lithiation may induce the same effects, and hence there is a reasonable pathway to a lithiated rocksalt by adding Li to V₂O₅.

In contrast, CoO₂ forms a layered structure: there is a very straight-forward pathway for lithiating CoO₂ by simply reducing Co and storing the lithium between the layers, but no plausible pathways for converting this structure to a rocksalt. Co₃O₄ forms a cubic spinel structure which has some space available between the tetrahedral and octahedral sites to incorporate Li, but it requires considerable atomic rearrangement to convert to a rocksalt-like structure. **Consequently, we do not expect the existence of a lithiated (or overlithiated) rocksalt phase in a Co-O system, unlike the existence of these phases in V-O and Nb-O systems.**

Furthermore, CoO is antiferromagnetic. This adds a considerable technical challenge in accurately simulating an overlithiated CoO rocksalt. Any calculation which ignores or otherwise incorrectly accounts for the magnetic moments in Co produces incorrect results. Substituting Co for Li at some sites in a rocksalt introduces significant complexities due to the magnetism of Co: should Li substitute at only spin-up (or spin-down) sites? Is it necessary to greatly expand the size of the unit cell so that pairs of Li can substitute both up- and down-spin sites? How does the distance between Li substitution sites influence magnetic coupling in the system? Or is a LiCoO₂ rocksalt even magnetic?

CoO₂ is also antiferromagnetic, but LiCoO₂ is non-magnetic. However, since Li substitutes between the planes in CoO₂, it is straightforward to add Li to antiferromagnetic CoO₂ and confirm that with the inclusion of Li the magnetic moments at the Co sites become negligible, and the solution converges to one in which magnetic effects were omitted from the outset (indeed, this is what we did).

In summary, we don't see any obvious chemistry or crystallography argument that suggests the existence of an overlithiated CoO rocksalt. Furthermore, because of the well-known significance of magnetic interactions in CoO, performing a reliable simulation of overlithiated CoO rocksalt is a much more significant undertaking than simply substituting Li for C, adding some interstitial Li to the rocksalt structure, and relaxing the geometry.

We therefore believe that a full investigation of overlithiated CoO rocksalt is well beyond the scope of the present paper and including a cursory comparison to a single naive overlithiated CoO rocksalt structure is of negligible value and likely to raise more questions than answers.

Regarding CoCO_3 , the referee is correct that surface carbonates frequently form in a wide variety of transition metal complexes, and there is probably some CoCO_3 in our systems as well. However, from a spectroscopy point of view, carbonate contamination is less of an issue in the higher transition metals than in the alkali and alkaline metals. This is because the energy of the spectroscopic features – in particular, the very sharp π^* orbital – is dominated by the CO_3 bonding rather than the nature of the transition metal. Consequently, these features tend to occur close to 534 eV, regardless of the system (compare the spectra from the alkali and alkaline carbonates to that of MnCO_3 from *The Canadian Mineralogist* 1995, 33 1157-1166).

In the large band gap alkali and alkaline oxides, the $\text{CO}_3 \pi^*$ orbital feature is at or below the pre-edge features from the native oxide. However, in the smaller band gap transition metal oxides, this feature is well above the pre-edge features of the native oxide. This is the case here: all of our cobalt oxides have pre-edge features near 531 eV and have no significant features in near 534 eV where we might expect the $\text{CO}_3 \pi^*$ orbital feature. In contrast, Li_2O has pre-edge features near 535 eV (as Li_2O has a large band gap). In fact, there is a small pre-edge feature at 534 eV in the measured data from Li_2O (which we have included in our revised draft), probably due to Li_2CO_3 contamination. The experimental and calculated O K-edge XANES of Li_2CO_3 are also added in the revised manuscript for comparison (**Figure S2 and Figure R1**).

Question 4: Since the authors are trying to compare measured XANES with computed XANES. It is very critical to benchmark the reliability of the functions they use. As shown in Figure S2, the peak position is very sensitive to functionals. The authors should compare their computed XANES with experimental measured XANES for at least some of the well-known materials such as CoO₂, Co₃O₄, Li₂O, CoO, etc.

Answer 4: Thank you for your constructive suggests. We have added the experimental O K-edge XANES of CoO, Li₂O, Co₃O₄, LiCoO₂ and Li₂CO₃ in the revised manuscript to compared that of experimental discharge/over-discharged electrode, as seen in Fig. 1 (revised manuscript), Fig. R1 and Fig. S2 (supportting information). In addition, since Li₂CoO₂ and Li_{1+x}CoO_{2-y} has no real object as a reference, we can only provide the calculated O K-edge spectrum as a reference. Here, we also calculated the O spectrum of CoO, Li₂O, Co₃O₄, LiCoO₂ and Li₂CO₃ materials for comparison with the experimental spectrum. We have performed the fitting with better reference spectra. The linear combination fit of D-0.0V and D-3.0V samples by experimental CoO, Li₂O, Co₃O₄ and LiCoO₂, and calculated Li₂CoO₂ are displayed in Figure R3. The linear combination fits of the experimental CoO, Li₂O, Co₃O₄, LiCoO₂ and calculated Li₂CoO₂ components to the O K-edge XANES spectra of D-3.0V and D-0.0V samples, allowing only energy shifting and intensity scaling of each component while keeping the overall spectral shape, yield the best fitting correspondence. The fit yielded LiCoO₂, Co₃O₄ and CoO contributions of 77.3, 14.9 and 7.8% to the O K-edge XANES of D-3.0V, respectively. The fit yielded LiCoO₂, Co₃O₄, CoO, Li₂O and calculated Li₂CoO₂ contributions of 8.1, 22.9, 21.9, 29.3 and 17.7% to the O K-edge XANES of D-0.0V, respectively. Necessary explanations and modifications have been made in revised manuscript, which shown in Fig. 1c and Fig. R3.

Question 5: It shows in Figure S3 that computed XANES for Li₁₃Co₁₁O₂₃ is very sensitive to ionic ordering. It is probably also sensitive to stoichiometry. In order to justify Li_{1+x}CoO_{2-y} can never show XANES that is similar to what has been observed at 0.0 V. More systematic evidence should be provided.

Answer 5: Thanks for your comments. Previous reports, like reference 32 and 33, often used the $\text{Li}_{1+x}\text{CoO}_{2-y}$ model to explain the phenomenon of overlithiation of LiCoO_2 . Here, we mainly use this model to rule out the possibility of overdischarge forming an oxygen vacancy model. If oxygen vacancies are generated, it may lead to severe distortion of the oxygen spectrum around the vacancies (Figure R11). We have related statements in the supporting information, which will lead to a small peak appears before 529 eV of O K-edge XAS, which is weakly related to the valence state of $\text{Co}^{2+/3+}$, as it does not appear in the oxygen spectrum of either calculated or actual CoO and Co_3O_4 (containing Co^{2+} ions). The calculated $\text{Li}_{1+x}\text{CoO}_{2-y}$ spectrum is different from the experiment one therefore we excluded this oxygen vacancy model.

Furthermore, XANES is dominated by short-range structure because the transition involves the overlap integral of the conduction band states with the core electron's wavefunction (which is obviously highly localized to the absorbing atom). In the present situation, our $\text{Li}_{13}\text{Co}_{11}\text{O}_{23}$ structure has some crystal symmetry (space group Bm) and results in 17 unique oxygen sites. We can break our simulation down into the XANES contribution from each of these sites. Most of these sites (12 out of the 17) provide XANES that are basically identical to that of LiCoO_2 . The three unique oxygen sites that bond directly to the substituted Li_{3a} all have the low-energy feature at 529 eV. The remaining two are bonded to Co but next to V_O , and these have additional fine structure in the main feature at 530 eV, which gets washed out when the contribution from all sites is averaged.

Consequently, our approach is less dependent on stoichiometry (and crystal structure) than it might seem. A full DFT study of a system with vacancies or impurities may require investigating a range of stoichiometries, as well as a range of geometries for each stoichiometry. However, if we are only focused on calculating XANES for structures with vacancies or impurities it is necessary to obtain only a structure with a reasonable concentration of these vacancies or impurities, and with a geometry that does not provide extreme clustering of these vacancies or impurities (i.e., in the present case, a Li_xCoO_y structure with an entire CoO_2 layer replaced with Li substituents and O vacancies is obviously extreme).

When the result shows that most of the site-specific spectra match those of the stoichiometric system (as is the case here), we can be reasonably confident that there is negligible interaction in the XANES spectra between the introduced defects. Consequently, we can construct a “background” XANES from all sites that match the stoichiometric system, and a “Li_{3a}+V_O” XANES from the specific sites that show distortion, and then linearly rescale the intensities these simulated XANES to investigate other stoichiometries (again, as long as the concentration of vacancies or impurities does not become so large that strong interactions between adjacent defects will have a significant effect). We do not quantitatively investigate this in the present paper because it is clear that in order to reproduce the measured spectra the “Li_{3a}+V_O” contribution to the XANES would need to be scaled to zero, as there is no feature in the measured data near 529 eV, and no new features from the “Li_{3a}+V_O” contribution new 538 eV (where the measured data most strongly differs from the spectra from stoichiometric LiCoO₂).

Figure R11 The calculated O k-edge XANES and geometrical configurations (inset) of Li_{1+x}CoO_{2-y}.

y.

Question 6: While the authors have much better evidence about Co₃O₄ existence, I have concerned that the evidence that Li₂CoO₂ is also formed is very weak. It should also be mentioned that the forming of spinel M₃O₄-like structure is very common in both LiCoO₂ and LiNiO₂, which can come from densification and Li loss at the surface. Surface reduction is very common in cathode materials, particularly after cycling. Thus the existence of Co₃O₄ at the surface can be totally irrelevant to overlithiation. The authors really want to have more convincing evidence to support that.

Answer 6: Thanks for your constructive comments. We made a linear combination fitting of O K-edge XANES of D-3.0V and D-0.0V electrodes. The results indicate the surface structure transformation of the LiCoO₂ electrode during overdischarge process (structure transition from LiCoO₂+Li⁺ to Co₃O₄, CoO, Li₂O and Li₂CoO₂). **Based on the LCF analysis, the quantitative results on the contribution of each component were obtained, as we discussed in Figure R3.** In addition, the XPS, HRTEM and in-situ sXRD analyses were also performed to investigate the surface chemical/structure evolution of LiCoO₂ electrodes during overdischarge, and the results are shown in Figures R4, R5 and R6 **(As we discussed in Reviewer #1 Question 4.** These discussions are also added in the revised manuscript and supporting information). The XPS spectra of D-3.0V and D-0.0V electrodes were used to detect the changes in chemical compositions in Figure R4. **Compared with that of the D-3.0V electrode (29.2% for Co²⁺), a large amount of Co²⁺ (62.8%) was found on the surface of the D-0.0V electrode, suggesting the surface Co reduction during overdischarge (Reduction to low-valent cobalt oxides after overdischarge, such as Li₂CoO₂/CoO/Co₃O₄).** These results are highly consistent with the conclusions of XANES spectra. **The HRTEM and corresponding FT/IFT images in Figure R5 clearly indicated that the existence of the cubic CoO/Co₃O₄/Li₂O phase.** Further analysis is performed on the 4 regions in Figure R5a: R1 represents the outer layered structure; R2 represents the inner layered structure; R3 represents the Co₃O₄ region; R4 represents the CoO region. The lattice fringes of a representative layer structure with a d-spacing of 4.90 Å in R1 and 4.76 Å in R2 could be assigned to the (003) plane of layered LiCoO₂. **The d₍₀₀₃₎-spacing in outer layered structure R1 region is greater than that in the R2 region, which may be due to the expansion of the interlayer spacing caused by the intercalation of an over-stoichiometric Li⁺ into the Li layer during the overlithiation process (the layered LiCoO₂ transformed to the layered Li₂CoO₂).** The lattice fringe with a d-spacing of 2.86 Å in R3 region could be assigned to the (220) plane of spinel Co₃O₄ (JCPDS file nos. 43–1003, a = 8.084 Å). It was worth noting that the R4 region exhibited a clear lattice fringe with a d-spacing of 2.13 Å, corresponding to the (200) plane of CoO (JCPDS file nos. 78–0431, a = 4.2667 Å).

(Nat. Commun. 2015, 6. Doi:10.1038/ncomms9106) In addition, for the surface of the D-0.0V electrode a thin coating with thickness 2-4 nm was bound to the surface of the LiCoO₂ structure and distributed in regions in Figure R5b. It was confirmed from the enlarged area and the FT image that the outer layer in Figure 5b is Li₂O phase, in which a d-spacing of 0.263 nm correspond to the (111) plane of Li₂O. **(Energy Storage Materials 2022, 51, 306-316.)**

Furthermore, in-situ synchrotron XRD of a LiCoO₂/graphite pouch cell was also used to investigate the structural evolution of the LiCoO₂ electrode during overdischarge from 3.0 to 0.0 V at 20 mA g⁻¹, and the result is shown in **Figure R6**. All the diffraction peaks have no significant change (the position of the main peak) and no impurity peaks appear (such as CoO/Co₃O₄/Li₂O phase), only a slight difference in intensity, which indicates that overlithiation is only a surface phenomenon and has little effect on the bulk structure. However, a small enhancement of (006) peak intensity of the LiCoO₂ phase can be observed during the overdischarge process, while the (012) peak intensity has no intensity change (insets of **Figures R6a and R6b**). All synchrotron XRD data were collected under the same conditions, so the intensity change in the diffraction peak could only be due to introduction of atom(s) in the unit cell, which resulted in a change in the *structure factor*. **So, the (006) peak increase may indicate excessive intercalation of Li⁺ in the Li layer, which in turn produces the Li₂CoO₂ structure, while the influence on other directions is small.**

Regarding Co₃O₄, the referee is correct that the forming of spinel M₃O₄-like structure is very common in both LiCoO₂ and LiNiO₂, especially in a highly delithiated state. **However, after careful analysis and discussion, we believe that the formation of the spinel Co₃O₄-like phase on the surface of LiCoO₂ electrode also occurs in the case of overlithiation.** Firstly, surface-sensitive XPS shows that the Co is heavily reduced on the surface of the D-0.0V electrode, compared to that of the D-3.0V electrode, which **is highly consistent with the conclusions of XANES spectra. The reduced Co during overdischarge can be considered to be caused by the formation of low-valence cobalt oxides, such as the formation of Li₂CoO₂/CoO/Co₃O₄-like phase.** As we discussed above, the formation of Li₂CoO₂/Li₂O/Co₃O₄/CoO-like phases

upon overlithiation of LiCoO₂ (D-0.0V electrode) has been confirmed by XANES, HRTEM, Raman and sXRD, together with spectrum simulations. Hence, the overdischarging reaction can be described by the following equations: $5\text{LiCoO}_2 + 3\text{Li}^+ + 3\text{e}^- \rightarrow \text{Li}_2\text{CoO}_2 + \text{Co}_3\text{O}_4 + \text{CoO} + 3\text{Li}_2\text{O}$. This means that the content of Co²⁺ occupies 60% in the overlithiated electrode, which is similar to the results of 62.8% for XPS analysis and 66.8% for XANES simulation.

In addition, the intensity of the Raman mapping of Co₃O₄ on LiCoO₂ particles of D-0.0V electrode (the area marked by the white dashed line in the Figure R12e) is significantly enhanced compared to that of the D-3.0V electrode (Figure R12b). These are also observed on the Raman spectrum in Figure R12c and R12f. Compared with the D-3.0V electrode, the Co₃O₄-like phase can be clearly observed at point 3' and point 4' of the D-0.0V electrode. This phenomenon is consistent with the notably Co₃O₄-like phase Raman peaks clearly observed on overdischarged LiCoO₂ mentioned in the observations in the literature. (The Journal of Physical Chemistry C 2010, 114, 3323-3328.) **Thus, based on the above discussion, we believe that these evidences can prove the formation of the spinel Co₃O₄-like phase on the surface of LiCoO₂ electrode also occurs in the case of overlithiation**

Figure R12 X-PEEM and Raman chemical imaging of discharged LiCoO₂ electrodes. **a, d** X-PEEM Co chemical mapping based on fitting Co L₃-edge image stacks (illustrations are Co L₃-edge XANES of red and green regions) of the electrode at two SOC states. **b, e** Bright-field reflection image (right) and Raman mapping of Co₃O₄ (687 cm⁻¹ peak intensity) and LiCoO₂ (A_{1g} peak intensity) (left) of the electrode at two SOC states. **c, f** Raman spectra extracted at selected spots on **(b** and

d). (a, b and c) are for the D-3.0V electrode; (d, e and f) are for the D-0.0V electrode.

Question 7: Some DFT details are missing from Figure 4. Are these computations done by GGA? GGA is well known to have the wrong prediction of electronic structure in Co-based layered materials. Hybrid functional should be a better way to quantify the bandgap.

Answer 7: Thanks for your comments. The details of the DFT calculation are found in the supporting information. All calculations were performed with with GGA+ U , using $U = 3.32$ eV for Co 3d orbitals. This value of U was selected to be consistent with previous literature (Nat. Mater. **18**, 496–502 (2019). <https://doi.org/10.1038/s41563-019-0318-3>). This approach has been demonstrated to produce accurate predictions of electronic structure in a wide variety of lithium transition metal oxides.

(Incidentally, LiCoO₂ is a system that seems to be surprisingly insensitive to the choice of functional – see *J. Mater. Chem. A* **2**, 107–115 (2014) <https://doi.org/10.1039/C3TA13387G> and *Surf. Sci.* **659**, 46–55 (2016) <https://doi.org/10.1016/j.susc.2016.01.004>).

To obtain accurate electronic structures, we added theoretical calculations using the Heyd-Scuseria-Ernzerhof (HSE06) screened hybrid functional, in which an amount of exact Hartree-Fock exchange mixing parameter of 0.25 is used, and the results are shown in Figure S12. We show that the HSE06 functional yields larger band gaps for LiCoO₂ and Li₂CoO₂ as compared to GGA+ U because it can eliminate self-interaction error, resulting in an over-delocalization of the electron. The hybrid functional calculation result of the band gap for LiCoO₂ is in good agreement with the literature (Physical Review B 2015, 92, 115118). However, as expected, the theory level impacts the band gap amplitude but not the global shape of the electronic band structures. These discussions are also added to the revised manuscript and supporting information, as shown in Figure R13.

Figure R13 The total density of states (DOS) and the partial density of states (pDOS) of LiCoO_2 (a) and Li_2CoO_2 (c); Computed Co-3d and O-2p pDOS of LiCoO_2 (b) and Li_2CoO_2 (d).

Question 8: In addition to the comment above, it is not convincing to claim that 0.079 eV of bandgap increase will lead to worse electronic conductivity. Electronic conductivity is related to both carrier concentration and mobility, the bandgap only controls the carrier concentration, while it is possible that Li_2CoO_2 will have better electronic mobility.

Answer 8: Thanks for your comments. Obtaining carrier mobility from first-principles calculations is not trivial, but it is almost certain that Li_2CoO_2 will have a lower carrier mobility than LiCoO_2 . Careful inspection of the DOS shows that Li_2CoO_2 has a much steeper slope at the VBM and CBM than LiCoO_2 , indicating the band structure is much flatter. Flatter bands mean higher effective masses, which generally lowers the carrier mobility. Unless Li_2CoO_2 has a proportionally larger mean free path (or mean free time between scattering), the carrier mobility in Li_2CoO_2 will be lower than that of LiCoO_2 . As Li_2CoO_2 has basically the same crystal structure as LiCoO_2 , we think it very unlikely that Li_2CoO_2 will somehow have a significantly larger mean free path than LiCoO_2 .

Incidentally, the CBM of Li_2CoO_2 is also dominated by Co 3d states with much weaker hybridization with Li/O states than LiCoO_2 , suggesting that excited electrons in Li_2CoO_2 are more likely to be correlated to Co sites and intrinsically less mobile than in LiCoO_2 .

In addition, according to the newly added HES06 calculation results, a bandgap increase of 0.307 eV is presented between LiCoO_2 and Li_2CoO_2 .

Question 9: It appears to me that the performance degradation of LiCoO_2 is irrelevant to the bulk property of formed Li_2CoO_2 and Co_3O_4 . Or at least, is not well supported by the data. In addition to the almost neglected difference in electronic conductivity between LiCoO_2 and Li_2CoO_2 and the mysterious diffusion barrier as mentioned above. Co_3O_4 spinel structure with a bit of off-stoichiometry will be a perfect framework for Li diffusion, which should be similar to the high rate spinel-based Li-Mn-O spinel cathodes. I am not sure if there is a reason that the authors did not show any DFT results in the Co_3O_4 phase.

Answer 9: Thanks for your constructive comments. Based on the above discussion, we show the difference in electronic conductivity between LiCoO_2 and Li_2CoO_2 . On the other hand, the lithiation of Co_3O_4 is fundamentally different from that of spinel LiMn_2O_4 . Spinel LiMn_2O_4 is the dominant insertion reaction, whereas the lithiation of Co_3O_4 accompanies the alloying or conversion reactions and may even contribute to spin-polarized capacitance (Nature Materials 2021, 20, 76–83). These alloying, conversion reactions and magnetoreception lead to the difficulty in establishing the structure change law, which makes it difficult to track the diffusion behavior of lithium in Co_3O_4 dynamic lithiation reaction from both experimental and theoretical calculations. Therefore, we did not use DFT calculation in this work to study effect of the Co_3O_4 lithiation process on over-lithiated $\text{Li}_{1+x}\text{CoO}_2$.

Question 10: The way the activation barrier is calculated is somehow inconsistent with the literature. As indicated by Fig. 4b, the $\text{Li01} \rightarrow \text{Li02}$ diffusion follows the oxygen dumbbell hop, which is demonstrated in previous papers to be unfavorable and unlikely to happen in LiCoO_2 . Somehow the authors demonstrated a much lower activation barrier than the literature, even without the appearance of divacancy. The authors should really check if their calculations are wrong, or at least wrote some comments to

compare their results with existing literature such as Journal of Power Sources 97-98 (2001) 529-531.

Answer 10: Thanks for your comments. Indeed, the diffusion behavior of layered intercalation compounds has long been discussed, and two mechanisms are generally considered: oxygen dumbbell hopping (ODH) and tetrahedral site hopping (TSH). Previous studies have shown that Li-ions tend to choose ODH at the early stage of charging (delithiation), and TSH begins to dominate when more than 1/3 Li-ions are extracted (J. Am. Chem. Soc. 2015, 137, 8364–8367). The literature presented by the reviewer (Journal of Power Sources 2001, 97–98, 529–531) also noted that the TSH mechanism requires the presence of a divacancy which becomes increasingly less likely at high lithium concentrations. In our calculations, we are initially based on the isolated vacancy model ($\text{Li}_{26}\text{Co}_{27}\text{O}_{54}$) and adopt the ODH mechanism to study the lithium diffusion barrier. Note that the calculated diffusion barrier (884 meV) is well consistent with the literature presented by the reviewer, as shown in Figure R15 (Journal of Power Sources 2001, 97-98, 529-531).

In addition, with the introduction of divacancy, the diffusion barrier decreases to 590 meV. Such reduction trend of the diffusion barrier is consistent with the previous report (Journal of Power Sources 2001, 97-98, 529-531), and the above values are within a reasonable range. These values depend on the strain effect of Li_xCoO_2 host and the change in effective valence of Co with x.

Figure R14. First-principles activation barriers for the TSH (squares) and ODH (circles) mechanisms at several different lithium concentrations x as calculated from first-principles. Note

that several different activation barriers can be calculated at a given lithium concentration since activation barriers depend on the local arrangement of lithium ions and vacancies around the migrating lithium ion, and many different such arrangements exist at fixed lithium concentration. (*Journal of Power Sources* 97-98 (2001) 529-531.)

Editorial Note: Figure above reprinted from *Journal of Power Sources*, **Volumes 97–98**, A. Van der Ven & G. Ceder, Lithium diffusion mechanisms in layered intercalation compounds, 529-531, Copyright (2001), with permission from Elsevier.

REVIEWERS' COMMENTS

Reviewer #1 (Remarks to the Author):

The reviewer appreciates the authors' effort in additional experiments, analyses, and responses to the review comments. The rigorousness of the science presented in this manuscript has been significantly improved with the new data provided. The reviewer agrees that this work will be an interesting and important contribution to the field and recommends the paper be published on Nat Commun. Just one comment concerning the styling: please increase the font sizes in the figures. For example, the d-spacings in Fig 1 are so small that after file compression they are almost indistinguishable.

Reviewer #3 (Remarks to the Author):

The authors have responded carefully to all my previous questions. I think the paper is greatly improved and will be appropriate to be published in Nature Communications

Reviewer #1:

Comments: The reviewer appreciates the authors' effort in additional experiments, analyses, and responses to the review comments. The rigorousness of the science presented in this manuscript has been significantly improved with the new data provided. The reviewer agrees that this work will be an interesting and important contribution to the field and recommends the paper be published on Nat Commun. Just one comment concerning the styling: please increase the font sizes in the figures. For example, the d-spacings in Fig 1 are so small that after file compression they are almost indistinguishable.

Respond: Thanks very much for your comments and your recognition of our work.

We have reorganized the images to improve the image quality, see Figure R1 for details.

Figure R1 XANES and HRTEM characterization of LiCo₂ electrodes. a The charge-discharge plot of LiCo₂, with the inset showing the expanded voltage range from 2.60 to 2.90 V; b Co L₃-edge XANES spectra of CoO, D-0.0V, D-3.0V, and LiCo₂ samples. c Linear combination fit of O K-edge XANES of D-3.0V and D-0.0V samples using experimental CoO, Li₂O, Co₃O₄, and LiCo₂ spectra, and the calculated Li₂Co₂ spectrum. d Comparison of the fitting components between D-3.0V and D-0.0V samples. e The calculated O K-edge XANES and the geometrical configuration (inset) of Li_{1+x}Co_{2-y}. f, g The geometrical configurations of LiCo₂ and Li₂Co₂. h HRTEM, corresponding FT and IFT images (R1, R2, R3 and R4 region) of D-0.0V electrode.

Reviewer #3:

Comments: The authors have responded carefully to all my previous questions. I think the paper is greatly improved and will be appropriate to be published in Nature Communications.

Respond: **Thanks very much for your comments and your recognition of our work.**